# AIR: Post-training Data Selection for Reasoning via Attention Head Influence

Jinrui Liu [*1]   Kai Hua [*2]   Xuanguang Pan [1]   Ge Zhang [3]   Yong Wang [2]   Shuai Ma [1]   Chongyang Tao [1]

## Abstract

LLMs achieve remarkable multi-step reasoning capabilities, yet effectively transferring these skills via post-training remains challenging. Existing data selection methods, ranging from manual curation to surface-level proxy heuristics based on length, entropy, or overall loss, fail to capture the causal importance of individual reasoning steps, limiting post-training efficiency. To address this, we propose Attention Influence for Reasoning (AIR), a principled, unsupervised and training-free framework that leverages mechanistic insights of the retrieval head to select high-value post-training data. AIR first identifies reasoning-critical attention heads of an off-the-shelf model, then constructs a weakened reference model with disabled head influence, and finally quantifies the resulting loss divergence as the Attention Influence Score. This score enables fine-grained assessment at both the step and sample levels, supporting step-level weighted fine-tuning and global sample selection. Experiments across multiple reasoning benchmarks show that AIR consistently improves reasoning accuracy, surpassing heuristic baselines and effectively isolating the most critical steps and samples. Our work establishes a mechanism-driven, data-efficient approach for reasoning distillation in LLMs.

## 1. Introduction

Large Language Models (LLMs) have demonstrated revolutionary capabilities in solving complex problems through multi-step chain-of-thought (CoT) reasoning (Guo et al., 2025; Yang et al., 2025). The ability to generate high-quality reasoning traces is critical, making research on improving and transferring these skills a primary focus in advanced generative AI. Seminal work in this field has primarily explored post-training techniques, including reasoning distillation via Supervised Fine-Tuning (SFT) (Muennighoff et al., 2025; Hugging Face, 2025; Guha et al., 2025; Hu et al., 2025; Ye et al., 2025) and approaches based on Reinforcement Learning (RL) (Schulman et al., 2017; Ziegler et al., 2019; Ouyang et al., 2022; Guo et al., 2025). While both paradigms have achieved notable progress in this area, RL is notoriously resource-consuming and difficult to tune, often rendering it prohibitively expensive. Fortunately, recent findings (Muennighoff et al., 2025; Hu et al., 2025) indicate that applying SFT to strategically curated, high-quality samples can match or even surpass the performance of more costly RL-based methods. This establishes a strong foundation where data-efficient and high-quality post-training is the key to democratizing advanced reasoning intelligence.

Given the demonstrated efficacy of post-training on limited examples, *selecting the most effective and high-value reasoning traces* has become paramount for maximizing knowledge transfer and enhancing reasoning capability. The community has explored various approaches: while manual curation, such as the hand-crafted sample selection performed by s1K (Muennighoff et al., 2025) and LIMO (Ye et al., 2025), shows strong performance, it is inherently labor-intensive and non-scalable. Automated solutions, including surface-level proxy heuristic methods based on length (Olsson et al., 2022) or complexity (Ye et al., 2025; Li et al., 2025; Wang et al., 2024) and influence-based scoring techniques (Lin et al., 2024a; Humane et al., 2025; Jiang et al., 2026; Qin et al., 2025), offer scalability. However, a significant gap remains: these existing methods rely on coarse proxy metrics which fail to isolate and measure the causal criticality of individual steps within a complex reasoning trace. This limitation becomes especially pronounced when reasoning relies on precise internal factual retrieval, which is performed by specialized mechanisms within Transformer architectures.

To bridge this gap, we advocate for principled data selection driven by insights from mechanistic interpretability (Olsson et al., 2022; Bricken et al., 2023a). We specifically leverage the finding that certain "retrieval heads" are functionally responsible for ensuring token-level copying and fidelity. Empirical studies demonstrate that masking these heads causes the model's multi-step reasoning capabilities to collapse, as it fails to refer back to key information (Wu

*Equal contribution [1]Beihang University [2]Independent Researcher [3]Bytedance. Correspondence to: Chongyang Tao <chongyangtao@gmail.com>.

*Proceedings of the 43rd International Conference on Machine Learning*, Seoul, South Korea. PMLR 306, 2026. Copyright 2026 by the author(s).

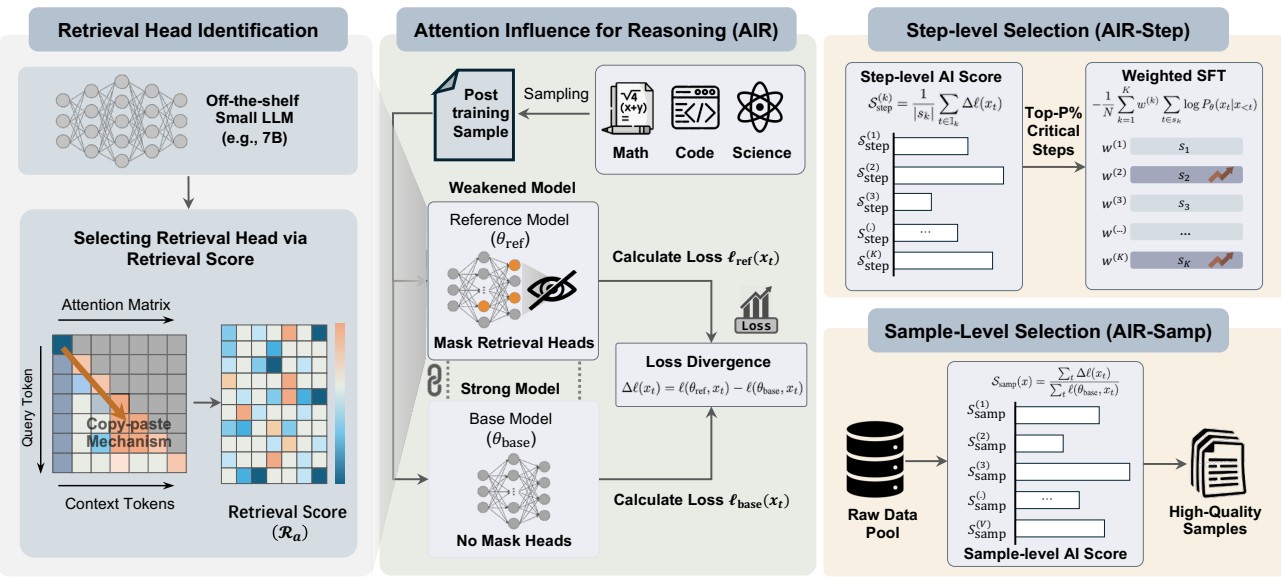

*Figure 1.* The illustration of the AIR method. Note that the base model and reference model are derived from the same underlying model. For ease of visualization, orange circles depict the masking of retrieval heads.

et al., 2025; Hua et al., 2025). Consequently, these heads serve as the backbone of multi-step reasoning, making them indispensable for the correct execution of reasoning steps. Building on this observation, we propose Attention Influence for Reasoning (AIR), an unsupervised and training-free framework for selecting post-training data based on the causal influence of reasoning-related attention heads. AIR first identifies reasoning-critical attention heads of an off-the-shelf base model ($\theta_{\text{base}}$). It then constructs a weakened reference model ($\theta_{\text{ref}}$) by disabling their influence, and measures the resulting loss divergence against the strong base model ($\theta_{\text{base}}$). This divergence directly yields the AIR score, a fine-grained metric reflecting the token's reliance on reasoning. We further compute the AIR Score at both the reasoning step-level and the sample-level, which are then used to drive two distinct data selection strategies: step-level weighted SFT and global sample selection.

We conducted comprehensive data selection experiments across both step-level and sample-level applications using several widely used reasoning datasets. The empirical results demonstrate that models distilled using our influence-based selection achieve notable improvements in accuracy, consistently outperforming conventional prioritization baselines. Furthermore, the effectiveness of our approach is supported by statistical analysis and visualization of the selected data distributions, which provide clear evidence that AIR successfully isolates and prioritizes the most critical steps and samples. The main contributions are threefold:

- We propose AIR, an unsupervised and training-free framework that quantifies the causal influence of reasoning-critical attention heads, enabling mechanism-driven data

selection for reasoning distillation.

- We formally define the AIR score as the loss divergence between a base model and its weakened counterpart obtained by masking retrieval heads, enabling step- and sample-level estimation of reasoning criticality.

- We comprehensively demonstrate that applying AIR leads to superior performance in transferring reasoning skills, outperforming existing selection methods and achieving performance comparable to manually curated data.

## 2. Related Work

**Data Selection for Pretraining.** Many training-free methods assess pretraining data quality using heuristic rules (Rae et al., 2021; Xie et al., 2023) or LLM perplexity (Ankner et al., 2025). For example, Scaling Filter (Li et al., 2024b) measures perplexity differences between small and large models on the same dataset. AttentionInfluence (Hua et al., 2025) is the first method to leverage transformer attention for fully unsupervised, training-free selection. Other work (Wettig et al., 2024; Zhao et al., 2024; Peng et al., 2025) trains multi-task classifiers using labels from commercial LLMs, with some combining domain-specific classifiers (Wettig et al., 2025) for practical applications.

**Data Selection for Post-training.** Unlike pretraining data selection (Yu et al., 2024), post-training selection works on smaller datasets, enabling more computationally expensive strategies. While manual curation guarantees quality (Muennighoff et al., 2025; Ye et al., 2025), it remains unscalable. Post-training samples can be assessed with richer posterior

information, for example by measuring the loss difference between the answer alone and the answer conditioned on the question (Li et al., 2024a), or by comparing the loss between answers given only the question and answers conditioned on both the question and its CoT (Wang et al., 2025a). Additional signals, such as question difficulty (Li et al., 2025), sample-level influence (Jiang et al., 2026) or token-level mirror influence (Lin et al., 2024a; Qin et al., 2025), can further guide the selection of informative samples or tokens for reasoning distillation. There is also another line of work (Wang et al., 2026; 2025b; Yu et al., 2026) that focuses on identifying the critical tokens or steps in reinforcement learning settings, providing insights into how token-level importance affects model learning and downstream reasoning.

**Mechanistic Interpretability** Understanding the inner workings of LLMs is crucial for advancing artificial general intelligence in a reliable and safe manner. Olsson et al. (2022) and Wu et al. (2025) reveal certain heads are responsible for in-context learning and retrieval, respectively. Lv et al. (2024) further explores how attention heads and MLPs collaborate for factual recall. Sparse autoencoders (Bricken et al., 2023b) and head importance estimation (Fu et al., 2024) are also used to analyze or optimize head behaviors. AIR adopts a proxy task, proposed by Wu et al. (2025); Qiu et al. (2024), to detect specific important heads, namely the retrieval heads in this paper. AIR naturally extends the insights from Wu et al. (2025), broadening their application beyond model analysis and inference acceleration to include effective and efficient data selection.

**Influence Measure** Ruis et al. (2025) uses influence functions to recognize pretraining documents important for learning factual knowledge and mathematical reasoning separately. Mirror Influence (Ko et al., 2024) realizes an efficient data influence estimation to select high-quality data. MATES (Yu et al., 2024) continuously adapts a data influence model to the evolving data preferences of the pretraining model and then selects the most effective data for the current pretraining progress. Our work is similar to Mirror Influence in that we use data influence estimation to select high-quality data. However, while Mirror Influence requires a high-quality dataset to train a strong reference model and create a model pair with significant differences in capabilities to compute the data-loss gap, our approach uses the attention mechanism to derive a weaker reference model from the base model. This enables us to obtain two models with a significant capability gap and compute the data-loss gap to evaluate data quality.

## 3. Method: AIR

We introduce AIR, a principled framework that employs model introspection based on attention influence to pro-

vide precise guidance for high-quality data selection for reasoning. Our core insight is derived from recent mechanistic interpretability findings: the existence of specialized retrieval attention heads is critical to LLMs' in-context learning, retrieval, and multi-step reasoning. As shown in Figure 1, we first construct a mechanically weakened reference model ($\theta_{\text{ref}}$) by specifically masking the identified reasoning-critical retrieval heads in the base model ($\theta_{\text{base}}$). Subsequently, we quantify data importance using the loss divergence between $\theta_{\text{ref}}$ and $\theta_{\text{base}}$—a metric that directly measures a sample's dependence on the model's fundamental retrieval mechanism. This approach yields the attention influence score ($\mathcal{S}$), which is applied at both the sample level (for prioritized data selection) and the reasoning step level (for fine-grained weighted supervision). Unlike current post-training data selection methods, which primarily rely on heuristic filtering or comparing losses against a strong reference model, AIR provides a training-free, cost-effective, precise, and mechanism-driven approach for curating impactful data essential for efficient reasoning post-training.

### 3.1. Identifying Important Attention Heads

Leveraging insights from mechanistic interpretability, we focus on isolating the specific components within various models based on Transformer architecture that underpin its capacity for information retrieval and complex reasoning. Recent findings indicate that retrieval heads play a critical role in maintaining LLMs' factual integrity and are pivotal for KV cache compression (Fu et al., 2024; Zhang et al., 2025). Crucially, they are critical for multi-step reasoning by enabling the "refer back" mechanism required for logical continuity, while their intrinsic consistency across scales empowers small models to serve as efficient guides for larger ones (Wu et al., 2025; Hua et al., 2025). Building on these insights, we operationalize the formal identification criterion to identify these crucial heads with a small LLM (e.g., Qwen2.5-7B-instruct) for our framework.

Formally, a retrieval head $h$ serves to preserve contextual fidelity by enabling accurate, token-level transfer of information from the source text to the generated output. To quantify this behavior, a token-level recall rate based on attention head dynamics is defined. Specifically, consider the LLM at decoding step $t$. Let the initial input context (the prompt) be $\mathbf{x}_{1:n}$, the sequence of previously generated tokens be $\mathbf{w}_{1:t-1}$. The full input sequence visible to the current query is $\mathbf{x}_{1:n+t-1}$. The attention score vector produced by head $h$ for the query $w_{t-1}$ (targeting the generation of $w_t$) is $\mathbf{a}_t \in \mathbb{R}^{n+t-1}$. An attention head $h$ is defined to successfully perform a context retrieval, or *"copy-paste"* operation, from a target content set $\mathbf{k} \subseteq \mathbf{x}_{1:n}$ to generate $w_t$ if and only if the following two conditions are satisfied:

**Token Appearance (C1):** The token generated at step $t$, $w_t$,

exists within the target context subset **k**: $w_t \in \mathbf{k}$.

**Maximal Attention (C2):** The position $j^*$ corresponding to the generated token $w_t$ must receive the maximal attention weight across all visible positions in head $h$:

$$j^* = \arg \max_{j \in \{1,\dots,n+t-1\}} \mathbf{a}_t[j], \quad \text{such that } x_{j^*} = w_t.$$

**Retrieval Score ($\mathcal{R}_h$).** Let $\mathbf{g_h}$ denote the set containing all tokens copied and pasted by a given head $h$ (i.e., tokens satisfying both C1 and C2). $\mathcal{R}_h$ quantifies the head's efficacy in retrieving the corresponding content **k** from the context, modeling this capability as a recall rate:

$$\mathcal{R}_h = \frac{|\mathbf{g_h} \cap \mathbf{k}|}{|\mathbf{k}|} \tag{1}$$

Attention heads exhibiting a high $\mathcal{R}_h$ are classified as *reasoning-critical retrieval heads* and are subsequently targeted for construction of the weakened reference model.

### 3.2. Calculating Attention Influence Score

After identifying the reasoning-critical retrieval heads using the criterion above, we continue by quantifying their functional impact on model behavior. To this end, we construct a weakened reference model by selectively masking these heads, thereby noising the contribution of their information pathways. We then measure the resulting loss divergence between the weakened model and the strong base model ($\theta_{\text{base}}$). This divergence, computed in terms of cross-entropy loss, serves as the foundation for defining the Attention Influence Score ($\mathcal{S}$), which captures how strongly each token, reasoning step, or sample is affected by the masked reasoning mechanisms.

**Weakened Reference Model Construction ($\theta_{\text{ref}}$).** Let $\mathcal{H}$ be the set of all attention heads in the base model $\theta_{\text{base}}$. The set of reasoning-critical retrieval heads, $\mathcal{H}_{\text{critical}}$, is defined by selecting all attention heads whose Retrieval Score ($\mathcal{R}_h$) falls within the top $\delta \in (0,1)$ percentile of $\{\mathcal{R}_h \mid h \in \mathcal{H}\}$. We then generate the weakened reference model $\theta_{ref}$ by systematically disabling the computational function of the identified retrieval heads $\mathcal{H}_{\text{critical}}$. This is achieved through a *masking operation* during the forward pass. For any layer $l$ and identified head $h \in \mathcal{H}_{\text{critical}}$, the attention weights are set to a uniform distribution. Specifically, if the attention sequence length is $L$, the attention weight $a_{ij}$ for all tokens $j$ is set such that:

$$a_{ij} = \frac{1}{L}, \quad \text{for all } j \in \{1, \dots, N\} \tag{2}$$

This masking operation effectively nullifies the head's specialized retrieval capability, degrading the model's performance specifically in fact retrieval and reasoning while keeping all underlying weight parameters identical to $\theta_{base}$.

**Quantifying Attention Influence via Loss Divergence.** The weakened model $\theta_{ref}$ serves as a critical counterpart to the strong base model $\theta_{base}$. We leverage the difference in their cross-entropy losses to calculate the *Attention Influence score for Reasoning ($\mathcal{S}$)*, hereafter referred to as the AIR score. Let $\ell(\theta, x_t)$ denote the token-level cross-entropy loss for a given model $\theta$ predicting the token $x_t$. The fundamental measure of influence for a single token is the loss gap between the two models:

$$\Delta\ell(x_t) = \ell(\theta_{\text{ref}}, x_t) - \ell(\theta_{\text{base}}, x_t) \tag{3}$$

A positive loss gap ($\Delta\ell(x_t) > 0$) signifies that the base model's performance on token $x_t$ heavily relied on the functional retrieval mechanism. This loss gap is then aggregated to derive the step-level and sample-level AIR scores, as detailed in the following sections. Although attention influence is computed at the token level, we do not directly perform token-level selection. Reasoning is inherently structured and step-wise, and individual tokens do not constitute meaningful training units. Moreover, token-level divergence is often noisy and fails to reliably capture underlying reasoning dependencies. More importantly, token-level selection would fragment the reasoning chain and disrupt the semantic coherence required for learning multi-step logical patterns. By aggregating token-level influence into step-level and sample-level scores, we can obtain more stable, semantically aligned indicators of reasoning importance that better match the sequence-level nature of SFT objectives. Therefore, step-level weighting and sample-level data selection can provide a more faithful way.

### 3.3. Sample-Level AIR Score

Based on the token-level loss divergence $\Delta\ell(x_t)$ derived from the $\langle\theta_{\text{ref}}, \theta_{\text{base}}\rangle$ model pair, we first quantify the influence at a global reasoning sample level. We define the total sequence loss for a sample $x$ as $\mathcal{L}(\theta, x) = \sum_{t=1}^{N} \ell(\theta, x_t)$. Since intrinsic sample difficulty varies, using absolute loss differences can be sensitive to the sample's scale. To mitigate this, we define the sample-level AIR score as the *relative loss divergence* :

$$\mathcal{S}_{\text{sample}}(x) = \frac{\sum_t \Delta\ell(x_t)}{\sum_t \ell(\theta_{\text{base}}, x_t)} \tag{4}$$

Samples with higher $\mathcal{S}_{\text{sample}}$ reflect a stronger reliance on the retrieval mechanism for accurate generation, suggesting they require strict contextual grounding to avoid logical fractures in the reasoning chain and are therefore more reasoning-intensive. In our experiments, we select the top $V$ samples with the highest $\mathcal{S}_{\text{sample}}$ scores for fine-tuning. This score provides an important signal for global data curation. Comparisons of $\mathcal{S}_{\text{sample}}$ are typically performed only within individual domains or subsets (for instance, with or without CoT), since loss values cannot be meaningfully compared across domains or subsets.

## 3.4. Step-Level AIR Score

While the sample-level metric curates training data from a global perspective, the step-level metric focuses on fine-grained local importance. We segment the reasoning trajectory into $K$ discrete reasoning steps, $\{s_1, s_2, \ldots, s_K\}$, typically delimited by structural separators (e.g., line breaks). Let $\mathbb{I}_k$ denote the set of token indices belonging to step $s_k$. The step-level AIR score for the $k$-th step is defined as the average loss divergence across all tokens within that step:

$$\mathcal{S}_{\text{step}}^{(k)} = \frac{1}{|s_k|} \sum_{t \in \mathbb{I}_k} \Delta\ell(x_t) \tag{5}$$

where $|s_k|$ denotes the number of tokens (i.e., the size of the index set $\mathbb{I}_k$) in step $s_k$.

Based on $\mathcal{S}_{\text{step}}^{(k)}$, we adopt a threshold-based weighting strategy to prioritize steps highly dependent on the retrieval heads. We define the set of critical steps $\mathcal{K}_P$ as the top $P$ percent of steps exhibiting the highest scores:

$$\mathcal{K}_P = \{k \mid \mathcal{S}_{\text{step}}^{(k)} \in \text{Top } P\% \text{ of } \mathcal{S}_{\text{step}}\} \tag{6}$$

The raw weight $\tilde{w}^{(k)}$ for step $k$ is then assigned based on membership in the critical step set $\mathcal{K}_P$:

$$\tilde{w}^{(k)} = 1 + (\alpha - 1) \cdot \mathbb{I}_{k \in \mathcal{K}_P} \tag{7}$$

where $\alpha \geq 1$ is the weight amplification factor that determines the boost applied to critical steps. The term $\mathbb{I}_{k \in \mathcal{K}_P}$ is the indicator function which evaluates to 1 if the step index $k$ belongs to the set of critical steps $\mathcal{K}_P$ and 0 otherwise.

All tokens $t$ belonging to step $k$ (i.e., $t \in I_k$) are assigned the uniform raw weight $\tilde{w}_t = \tilde{w}^{(k)}$. We apply sequence-level normalization to prevent weight magnitude from biasing the global learning rate. For each post-training sample, which consists of a clearly defined model input and model output, we denote the total token count across all steps in the sample as $N$, namely the total number of tokens in the output. The final normalized weight $w^{(k)}$ is scaled such that the weighted sum equals the total token count $N$:

$$w^{(k)} = \tilde{w}^{(k)} \times \frac{N}{\sum_{j=1}^{K} |s_j| \cdot \tilde{w}^{(j)}} \tag{8}$$

Finally, the weighted SFT objective is applied as follows:

$$\mathcal{L}_{\text{SFT}} = -\frac{1}{N} \sum_{k=1}^{K} w^{(k)} \sum_{t \in s_k} \log P_\theta(x_t | x_{<t}) \tag{9}$$

# 4. Experiments

In this section, we present empirical analysis to validate the effectiveness of reasoning-intensive data selected by AIR.

## 4.1. Datasets and Metrics

Following the experimental protocol of s1 (Muennighoff et al., 2025), we utilize datasets derived from the s1 project[1]. For sample-level experiments, we apply AIR mechanism directly to the raw 59K-full dataset to identify high-value samples. Given that the 59K-full dataset contains only trajectories distilled from Gemini (Team et al., 2023), we construct a corresponding version distilled from DeepSeek-R1 for each selected example to validate performance variations across different reasoning trajectories[2]. For step-level experiments, given its superior performance, we employ s1K-1.1, an improved version of s1K that adopts reasoning trajectories generated by DeepSeek-R1 (Muennighoff et al., 2025). For evaluation, we select four mainstream benchmarks categorized into mathematical reasoning and general scientific capabilities, following the benchmark setup of Muennighoff et al. (2025). The mathematical domain includes AIME 2024 and 2025, which assess competition-level proficiency in algebra, geometry, and number theory, and MATH500, which evaluates comprehensive mathematical problem-solving capabilities. The scientific domain includes GPQA Diamond, consisting of 198 PhD-level problems across biology and chemistry. We conduct all evaluations using the official codebase provided by s1 (Muennighoff et al., 2025), utilizing greedy decoding, and we report Pass@1 accuracy. Further details on the datasets are provided in Appendix A.1.

## 4.2. Baselines

Sample-level experiments adopt the s1 series models as primary baselines. These models are trained on a 1K high-quality subset selected from the 59K dataset with manual assistance, comprising the s1K model (based on Gemini trajectories) and the s1K-1.1 model (based on DeepSeek-R1 trajectories). The experiments also incorporate simple heuristic baselines, including Random, Length, and Diverse, along with three metric-based selection strategies: IFD (Li et al., 2024a), which measures instruction-following difficulty; Entropy (Wang et al., 2026), which quantifies model uncertainty; and RHO-sample (Lin et al., 2024a), which evaluates the performance discrepancy between a reference model and the current model. Step-level AIR experiments are conducted on the s1K-1.1 dataset, and thus we primarily compare against the s1K-1.1 model. Furthermore, we introduce three alternative weighting strategies: Random, which assigns random weights; Entropy (Wang et al., 2026), which utilizes the base model ($\theta_{\text{base}}$) to compute the average Shannon entropy within steps; and the RHO-step method. Finally, we specifically include DeepSeek-R1 and its dis-

---

[1] https://github.com/simplescaling/s1
[2] Due to cost constraints, we do not create the complete 59K R1-reasoning dataset.

*Table 1.* Performance of AIR-sample and AIR-step . The symbol † indicates the involvement of humans.

| Model | # Examples | AIME 2024 | AIME 2025 | MATH500 | GPQA Diam. | Average |
|---|---|---|---|---|---|---|
| R1 (Guo et al., 2025) | ≫800K | 79.80 | 70.00 | 97.30 | 71.50 | 79.65 |
| R1-distill-Qwen-14B (Guo et al., 2025) | 800K | 61.70 | 48.00 | 93.90 | 59.10 | 65.68 |
| R1-distill-Qwen-32B (Guo et al., 2025) | 800K | 58.30 | 49.60 | 94.30 | 62.10 | 66.08 |
| R1-distill-Llama-70B (Guo et al., 2025) | 800K | 57.10 | 56.30 | 94.50 | 65.20 | 68.28 |
| **s1K Setting: Distilling from Gemini** | | | | | | |
| Random 1K | 1K | 30.00 | 20.00 | 90.40 | 51.01 | 47.85 |
| Diverse 1K | 1K | 26.70 | 23.33 | 91.20 | 54.60 | 48.96 |
| Length 1K | 1K | 33.30 | 26.70 | 90.40 | 59.60 | 52.50 |
| IFD (Li et al., 2024a) | 1K | 30.00 | 23.33 | 89.80 | 52.52 | 48.91 |
| Entropy (Wang et al., 2026) | 1K | 30.00 | 26.70 | 88.80 | 54.04 | 49.89 |
| RHO-sample (Lin et al., 2024a) | 1K | 23.33 | 20.00 | 88.40 | 48.99 | 45.18 |
| s1K† (Muennighoff et al., 2025) | 1K | 50.00 | 26.70 | 92.60 | 56.60 | 56.48 |
| **AIR-Sample** | 1K | 50.00 | 23.33 | 90.80 | 55.00 | 54.78 |
| **s1K-1.1 Setting: Distilling from Deepseek R1** | | | | | | |
| Random 1K | 1K | 50.00 | 36.67 | 94.80 | 58.59 | 60.02 |
| s1K-1.1† (Muennighoff et al., 2025) | 1K | 56.70 | 50.00 | 94.40 | 60.60 | 65.43 |
| **AIR-Sample** | 1K | 56.70 | 50.00 | 95.20 | 66.67 | 67.14 |
| s1K-1.1 + Random | 1K | 56.70 | 50.00 | 94.00 | 61.62 | 65.58 |
| s1K-1.1 + RHO-Step (Lin et al., 2024a) | 1K | 66.70 | 46.67 | 94.40 | 65.66 | 68.36 |
| s1K-1.1 + Entropy (Wang et al., 2026) | 1K | 63.33 | 43.33 | 93.80 | 64.14 | 66.15 |
| **s1K-1.1 + AIR-Step** | 1K | 66.70 | 53.33 | 95.60 | 65.66 | 70.32 |

tilled models as stronger reference baselines (Guo et al., 2025). Detailed descriptions of the baselines are provided in Appendix A.2.

### 4.3. Implementation Details

We select Qwen2.5-7B-Instruct (Yang et al., 2024) as the base model for data selection and reference model construction. Specifically, driven by the finding in Wu et al. (2025) that retrieval heads consistently constitute approximately 5% of all heads across various models and scales, we construct the reference model $\theta_{\text{ref}}$ by masking the top 5% of attention heads ranked by Retrieval Score. For the post-training phase, we employ the Qwen2.5-32B-Instruct model and strictly follow the s1 experimental protocol (Muennighoff et al., 2025). In the step-level setting, we set the ratio of critical steps ($P\%$ in Equation 6) to 20%, selecting the steps with the highest Attention Influence Score, and set the weight amplification factor ($\alpha$ in Equation 7) to $\alpha = 2$. In the sample-level setting, to maintain a category distribution (e.g., Math, Science, and Crossword) consistent with the s1K/s1K-1.1 datasets, we allocate sampling quotas according to the corresponding category proportions and perform stratified sampling based on the $\mathcal{S}_{\text{samp}}$ score, thereby constructing a training set of comparable size ($V = 1000$). Additional experimental details are provided in Appendix A.3.

### 4.4. Evaluation Results

Table 1 presents the performance of different sample-level data selection methods (highlighted in purple) and step-level selection methods (highlighted in yellow).

***Sample-Level Performance.*** The results clearly demonstrate that our AIR-sample outperforms both heuristic baselines (e.g., Random, Length, and Diverse) and metric-based baselines (e.g., Entropy, IFD, and RHO-sample). By replacing random selection with our AIR-sample strategy, the average accuracy under the s1K setting increases from 47.85% to 54.78%, highlighting its value in choosing high-quality training samples. Among the baselines, excluding the manually curated s1K, Length is the most competitive, achieving the highest score on GPQA Diamond (59.60%), consistent with the findings of the s1 paper (Muennighoff et al., 2025). This is reasonable, as longer samples typically contain richer reasoning steps and more complex contexts, providing more informative training signals.

Compared with the *manually curated* s1K/s1K-1.1 baseline, our AIR-Sample achieves slightly lower or comparable performance to s1K, while consistently outperforming s1K-1.1 across all four datasets. Notably, our method relies solely on automatic AIR scores, eliminating the need for manual filtering, which demonstrates the efficiency and scalability of our approach. In addition, the superior performance of AIR on s1K-1.1, where responses are replaced with stronger reasoning trajectories from DeepSeek-R1, further confirms that high-quality problems selected by our strategy, when combined with stronger reasoning paths, can more fully unlock the model's potential. Significantly, under the s1K-1.1 setting, models trained on only 1K examples selected by AIR even outperform R1-distill-Qwen-32B, which is trained on 800K examples distilled from DeepSeek-R1, highlight-

*Table 2.* Impact of retrieval head masking and model scale. RandomH denotes computing influence scores by randomly masking 5% of attention heads.

| Method | AIME 2024 | MATH 500 | GPQA Dia. | AVG |
|---|---|---|---|---|
| *Sample-Level Setting* | | | | |
| s1K | 50.00 | 92.60 | 56.60 | 66.40 |
| Random 1K | 30.00 | 90.40 | 51.01 | 57.13 |
| RAIR-Sample (7B) | 26.67 | 90.00 | 51.01 | 55.89 |
| AIR-Sample (7B) | 50.00 | 90.80 | 55.00 | 65.27 |
| AIR-Sample (1.5B) | 40.00 | 88.20 | 56.06 | 61.42 |
| *Step-Level Setting* | | | | |
| s1K-1.1 | 56.70 | 94.40 | 60.60 | 70.57 |
| Random | 50.00 | 94.80 | 58.59 | 67.79 |
| RAIR-Step (7B) | 53.33 | 94.60 | 60.10 | 69.34 |
| AIR-Step (7B) | 66.67 | 95.60 | 65.66 | 75.98 |
| AIR-Step (1.5B) | 63.33 | 94.60 | 62.12 | 73.35 |

ing the effectiveness of our approach in leveraging small, high-value training subsets. We also analyze data processing efficiency in Appendix A.4.

***Step-Level Performance.*** Compared to the s1K-1.1 baseline (65.43% average accuracy), our AIR-Step substantially improves performance, raising the average accuracy to 70.32%. These results suggest that applying fine-grained loss weighting to reasoning steps allows the model to better capture critical reasoning logic, thereby enhancing both logical rigor and problem-solving proficiency on complex mathematical and scientific tasks. Besides, AIR-Step demonstrates highly competitive performance against other step-level strategies and even outperforms the supervised method RHO-step, despite not relying on supervision. Furthermore, using only 1,000 training samples with step-level weighted fine-tuning, our method outperforms the R1-distill-Qwen-32B model, which is trained on 800K examples. It also surpasses the R1-distill-Qwen-70B model on most benchmarks, highlighting its remarkable data efficiency.

### 4.5. Discussion

***Effects of Retrieval Heads in Reference Model.*** To investigate the pivotal role of retrieval heads and verify that our method does not benefit simply from the act of masking attention heads, we computed influence scores by randomly masking 5% of attention heads instead of relying on retrieval scores defined in Equation 1, which we term RAIR. As shown in Table 2, experimental results under both sample-level and step-level indicate that random masking fails to discern the intrinsic reasoning value of the data. In sample-level setting, the performance of RAIR degrades significantly, while in step-level weighting experiments, RAIR also fails to yield performance gains. This suggests that the effectiveness of AIR stems from leveraging mechanistic interpretability insights to precisely identify retrieval

heads critical for reasoning, rather than from the mere act of masking attention heads.

***Impact of Reference Model Scale.*** To explore the impact of reference model scaling on AIR, we evaluated the performance of sample-level selection and step-level weighting using a 1.5B reference model. As shown in Table 2, a positive correlation exists between model scale and data selection quality, with larger models proving more effective at identifying reasoning-intensive samples. Notably, the performance of sample-level selection degrades when the performance gap between the reference model and the model to be trained is substantial. However, the 1.5B model continues to demonstrate robust performance, surpassing the corresponding random baselines significantly in both sample-level and step-level experiments. This indicates that while smaller models can effectively locate key reasoning steps via the AIR mechanism, larger models are superior in recognizing complex reasoning-intensive patterns and providing more precise supervision signals.

***Hyperparameter Analysis.*** To evaluate the robustness of AIR and its dependence on hyperparameter settings, we conducted a sensitivity analysis on two key hyperparameters in the step-level AIR SFT process: the ratio of selected critical steps ($P\%$ in Equation (6)) and the weight amplification factor ($\alpha$ in Equation (7)). Figure 2 presents the evaluation results. We can observe that: as the ratio of selected critical steps increases, model performance first improves and then declines. When $P\%$ is set to 20%, the model achieves peak performance across two major benchmarks (AIME 2024 and GPQA). However, when the ratio is further increased to 40% or 50%, a substantial number of non-critical or redundant steps are included in the high-weight category, leading to diminished performance. Similarly, the weight amplification factor $\alpha$ follows a consistent pattern, with a multiplier of $\alpha = 2$ yielding the optimal overall gain. Overly high weights ($\alpha \geq 5$) may induce aggressive local gradient updates that distort the loss landscape, causing the model to overfit specific reasoning fragments at the expense of global coherence. Therefore, adopting a moderate weight multiplier of $\alpha = 2$ is proper for maintaining the model's general reasoning capabilities.

***Analysis of AIR Selected Samples.*** To comprehensively elucidate how different selection strategies shape the underlying data distribution and enhance post-hoc interpretability, we consider several evaluation metrics to systematically compare the dataset selected by AIR with the baseline s1K-1.1 dataset. Specifically, the framework assesses logical complexity through the *Average Reasoning Steps* (ARS), which quantifies the granularity and depth of the reasoning path. In addition, we examine *Prompt Constraint Density* (PCD) to capture the richness and specificity of restrictive conditions within input instructions. Finally, we evaluate

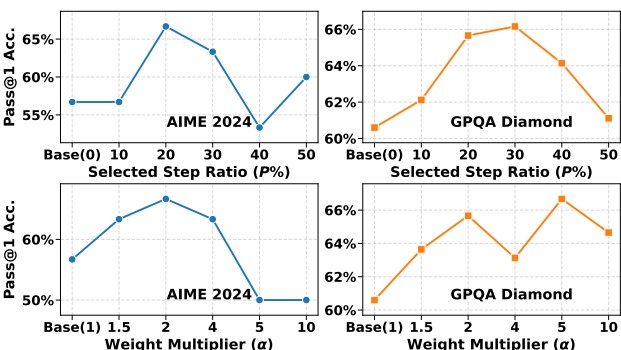

*Figure 2.* Hyperparameter Analysis in the step-level AIR SFT.

*Table 3.* Comparison of sample characteristics between s1K and AIR selected dataset.

| Metric | s1K-1.1 | AIR-Sample (Ours) |
|---|---|---|
| Average Reasoning Steps (ARS) | 265.15 | 274.05 |
| Prompt Constraint Density (PCD) | 1.11% | 1.45% |
| Numeric Answer Ratio (NAR) | 54.48% | 42.24% |
| Symbolic Answer Ratio (SAR) | 45.52% | 57.76% |

*Table 4.* Comparison of reasoning characteristics between s1K-1.1 and s1K-1.1+AIR-Step models on AIME 2024.

| Metric | s1K-1.1 | s1K-1.1 (+AIR-Step) |
|---|---|---|
| *Macro Statistics* | | |
| Avg. Response Length | 10,652.87 | 10,537.00 |
| Transition Density (%) | 8.846 | 8.862 |
| *Reasoning Connectives (%)* | | |
| Causal | 2.82 | 2.80 |
| Contrast | 1.05 | 1.13 |
| Correction | 1.06 | 1.16 |
| Exploratory | 1.54 | 1.54 |
| Planning | 0.97 | 1.04 |
| Verification | 1.02 | 1.08 |
| *Symbolic & Structure(%)* | | |
| Symbolic Ratio | 2.37 | 2.52 |
| Header Line | 0.03 | 0.28 |
| List Line | 9.97 | 9.64 |
| *Reasoning Rhythm* | | |
| Sentence Volatility (CV %) | 94.65 | 96.14 |

answer complexity using the *Numeric Answer Ratio* (NAR) and *Symbolic Answer Ratio* (SAR): defined as the proportions of purely numeric answers and answers containing symbolic expressions, respectively to characterize the extent to which solutions rely on formal logical deduction. For detailed metric definitions and computation procedures, please refer to Appendix A.5. The statistical results presented in Table 3 reveal the advantages of AIR dataset regarding cognitive load and logical density. In terms of reasoning characteristics, the *Average Reasoning Steps* of AIR (274.05) are higher than those of s1K-1.1 (265.15). This indicates that samples selected via attention influence tend to require more granular problem decomposition and more complex logic. Regarding cognitive constraints, AIR exhibits a higher Prompt Constraint Density (1.45% vs. 1.11%), implying that the selected problems often involve more specific or stringent boundary conditions, thereby requiring the model to perform highly consistent deductions within a given logical framework. Crucially, in terms of answer complexity, the proportion of answers containing symbolic and formulaic derivations in AIR reaches 57.76%, substantially surpassing the 45.52% observed in s1K-1.1. This structural disparity suggests that AIR prioritizes the retention of samples necessitating symbolic manipulation, abstract generalization, and structured deduction. Furthermore, we conduct a microscopic analysis of the logical connective distributions in Appendix B.1, and visualize the distributions of the selected dataset and the s1K-1.1 dataset using PCA for dimensionality reduction in Appendix B.2.

***Impact of AIR-Step on Reasoning Behavior.*** To investigate changes in reasoning behavior at a granular level, we con-

ducted a quantitative semantic analysis on the responses generated by the s1.1 baseline and our AIR model on the AIME 2024 benchmark.[3] Table 4 illustrates the differences in reasoning dynamics between the two models. At the macro level, results based on dependency syntactic analysis reveal that, despite a slight reduction in average response length, the Transition Density of the AIR model increased from 8.846% to 8.862%, indicating enhanced logical coherence. At the micro level, an analysis of logical connective distribution shows a marked rise in the density of "Correction," "Planning," and "Verification" tokens. This reflects more frequent reflection and self-correction during the reasoning process, demonstrating a more rigorous reasoning chain. Regarding structural organization, in contrast to the baseline's preference for linear lists (List Lines), the AIR model exhibits a significant increase in the usage of Header Lines and mathematical symbols. This shift indicates that the AIR strategy successfully encourages the model to decompose complex thought processes into modular logical units. Furthermore, the increased variance in sentence length mirrors a human-like reasoning rhythm which employs long sentences for complex deduction and short ones for qualitative confirmation, thereby facilitating the construction of a more robust logical framework. We also provide additional case studies including analyses of the model outputs (Appendix B.3).

***Case Study: Steps Selected by the AIR Model*** Figure 4 illustrates how the AIR model selects critical reasoning steps. Steps marked in red indicate those chosen by AIR, while some intermediate reasoning steps have been omitted for clarity. The AIR scoring mechanism emphasizes

---

[3]Details of the metrics are in Appendix B.4.

**Problem**: Given a rational number, write it as a fraction in lowest terms and calculate the product of the resulting numerator and denominator. For how many rational numbers between 0 and 1 will 20! be the resulting product?

**Reasoning Snippet**

...

Therefore, the number of such coprime pairs is $2^k$, where $k$ is the number of distinct prime factors of 20!. But wait, 20! has prime factors: 2, 3, 5, 7, 11, 13, 17, 19. That's 8 primes. Since all exponents except for 2, 3, 5, 7 have exponent 1, and exponents for 2, 3, 5, 7 are even and odd.

Therefore, 20! is not a perfect square. Thus, there are no solutions where $a = b$. Hence, total coprime pairs are $2^8$. But since each assignment corresponds to a unique pair, total number of unordered pairs would be $2^{8-1} = 128$.

Wait, perhaps not quite. Since not all assignments will satisfy $a < b$, even for coprime divisors. Wait, confusion arises here.

Wait, actually, we need to be precise. The assignments consist of selecting a subset $S$ of the set of prime factors. The number of pairs with $a < b$ is equal to the number of possible $a$ such that $a < \sqrt{20!}$. Since 20! is not a square, the total number should be $256/2 = 128$.

But wait, can it be? Let me check with a smaller factorial.

Suppose $n = 6$. $6! = 720$. Prime factorization: $2^4 \times 3^2 \times 5^1$. Primes are 2, 3, 5 (3 primes). Total assignments $2^3 = 8$. We expect $8/2 = 4$ pairs where $a < b$.

So for 720, with prime factors 2, 3, 5: The 8 assignments: 1. Assign none to a => a=1... 2. Assign 2 to a... ... 5. Assign 2 and 3 to a => a=144, b=5 (a > b: no)

Therefore, out of 8 assignments, 4 result in $a < b$. So in general, the answer seems to be half of the number of coprime ordered pairs. For 20!, it's $2^8/2 = 128$.

...

**The final answer: 128**

*Figure 4.* Visualization of critical reasoning steps selected by our AIR. The example illustrates a self-correction process: the model first derives an answer, then questions its logic with respect to the $a < b$ constraint, performs a verification on a smaller case (6!), and ultimately confirms the result. For ease of illustration, only key reasoning steps are shown.

*logical pivots* that guide the reasoning direction and key *self-correction* moments where the model revises incorrect assumptions. Routine declarations or low-information intermediate calculations are assigned lower weights. By focusing optimization on the most information-dense steps, AIR effectively distills the causal structures essential for maintaining coherent reasoning chains, leading to improved performance on complex multi-step tasks. More empirical cases are provided in Appendix G.

## 5. Conclusion

In this paper, we propose Attention Influence for Reasoning (AIR), an unsupervised and training-free framework for selecting high-quality and reasoning-intensive post-training data. Specifically, AIR first identifies "retrieval heads" responsible for token-level copying, it then constructs a weakened reference model by disabling their influence and quantifies the loss divergence relative to the strong base model to enable two approaches: step-level weighted supervised fine-tuning and global sample selection. Experiments across multiple reasoning and comprehensive benchmarks demonstrate that AIR identifies critical reasoning steps and samples, consistently improves reasoning performance, and establishing a mechanistically interpretable and data-efficient approach for LLM reasoning post-training.

## Acknowledgments

We thank the anonymous reviewers for their constructive comments and suggestions. This work was supported by the National Natural Science Foundation of China (Grant No. 62572034, U22B2021, and U24B20143), State Key Laboratory of Complex & Critical Software Environment (SKLCCSE), and CIE–Tianyi Cloud Research Program.

## Impact Statement

This paper presents work whose goal is to advance the field of Machine Learning. There are many potential societal consequences of our work, none which we feel must be specifically highlighted here.

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

# A. Experimental Setup

## A.1. Dataset Details

In this work, we leverage the datasets introduced by the s1 project (Muennighoff et al., 2025). The details of these datasets are described below:

**1. The 59K-full Dataset.** This corpus comprises approximately 59,000 questions compiled from 16 distinct sources. Furthermore, the pool incorporates novel, manually curated quantitative reasoning datasets, such as PhD-level statistics problems (s1-prob) and brain teasers (s1-teasers). A manual inspection was conducted during the collection phase to ensure data quality, resulting in the direct exclusion of datasets characterised by poor formatting or insufficient reasoning depth. We utilize this raw data pool for our sample-level filtering experiments.

**2. The s1K Dataset.** The s1K dataset is a high-quality subset of 1,000 samples derived from the 59K-full dataset.The curation follows four criteria: *Quality*, *Difficulty*, *Diversity*, and manual selection. To construct this dataset, Muennighoff et al. (2025) employed a pipeline that integrates automated heuristics with manual curation. The extraction procedure consists of the following stages:

**Stage 1: Quality Filtering.** The process initiates with an automated filter designed to remove samples that trigger API errors, as well as those containing specific string patterns indicative of formatting issues (e.g., ASCII art, malformed citations). This step reduced the data pool to approximately 51.5K samples.

**Stage 2: Difficulty Screening.** To guarantee sufficient problem complexity, a model-based difficulty filter was applied. Each question was evaluated using two models (Qwen2.5-7B-Instruct and Qwen2.5-32B-Instruct). Questions correctly solved by either model were deemed "too trivial" and subsequently discarded. This filtering step significantly reduced the pool to approximately 24.5K samples.

**Stage 3: Diversity Awareness.** This stage aims to ensure broad coverage and balanced complexity in the selected data by combining manual curation with domain-stratified sampling.

- **Manual Pre-selection (384 samples):** Based on manual quality assessments, 384 samples were directly selected from high-reliability datasets.
- **Diversity Sampling (616 samples):** The remaining quota was filled using a stratified sampling algorithm. Questions were categorized into 50 scientific domains (e.g., number theory, quantum mechanics) via the MSC system. The sampling algorithm ensured uniform domain coverage while prioritizing questions with longer reasoning trajectories, which served as a proxy for complexity.

**3. The s1K-1.1 Dataset** The s1K-1.1 dataset retains the identical set of 1,000 questions selected through the aforementioned pipeline but regenerates the reasoning trajectories using **DeepSeek-R1**. Compared to the Gemini-distilled trajectories utilized in the original s1K, the trajectories in s1K-1.1 are generally more extensive and demonstrate superior reasoning capabilities, thereby providing higher-quality supervision signals for our step-level experiments.

## A.2. Baselines

We compare AIR against several sample-level selection strategies:

- **Heuristic Baselines:** Following s1 (Muennighoff et al., 2025), we evaluate *Random* (random stratified sampling by task type), *Length* (favoring longer reasoning chains), and *Diverse* (clustering-based selection).

- **IFD Score (Li et al., 2024a):** Selects samples with the highest Instruction-Following Difficulty, defined as the ratio of the conditional loss to the direct loss.

- **Entropy (Sample-level):** Prioritizes samples with high average token entropy in the response, serving as a proxy for model uncertainty.

- **RHO-sample:** We implement the contrastive selection strategy proposed by Lin et al. (2024b). We employ the **Qwen-7B-Instruct** model fine-tuned for 3 epochs on the manually curated LIMO dataset (Ye et al., 2025) as the strong reference model, while utilizing the original model as the weak model. To ensure a balanced distribution, we apply

stratified sampling based on category proportions to construct the dataset, consistent with the selection strategy used in our main experiment.

For step-level weighting, we implement three comparative strategies:

- **Random Step:** Assigns high weights to 20% of steps randomly selected from each instance, ignoring semantic content.

- **Entropy (Step-level):** Uses the base model $\theta_{\text{base}}$ to compute the Shannon entropy for each token from the output logits. We calculate the mean entropy over all tokens within a step and select the top 20% of steps with the highest uncertainty for weighting, matching the selection ratio of AIR.

- **RHO-step:** Leveraging the same strong and weak model pair constructed in the RHO-sample setting, we compute the loss difference for each reasoning step. We then select the top 20% of steps with the highest contrastive scores for weighted fine-tuning, matching the selection ratio of AIR.

### A.3. Training Details

All training experiments are executed on a platform equipped with 8 NVIDIA A800 GPUs, utilizing DeepSpeed for distributed training. Our post-training implementation strictly adheres to the experimental protocol of s1 (Muennighoff et al., 2025). Models are fine-tuned using the AdamW optimizer (Loshchilov & Hutter, 2017) with parameters $\beta_1 = 0.9$, $\beta_2 = 0.95$, and a weight decay of $10^{-4}$. The learning rate is initialized at $10^{-5}$ and follows a cosine decay schedule after a linear warmup over the first 5% of training steps. Regarding context configuration, we adopt the settings from s1.1, setting the maximum sequence length for DeepSeek-R1-generated reasoning traces to 20,000 tokens. We mask the user prompt and calculate cross-entropy loss solely on the reasoning trajectories and final responses. Furthermore, to optimize memory efficiency, we employ bfloat16 precision throughout the training process.

### A.4. Efficiency in Data Selection with AIR

AIR demonstrates high efficiency in selecting high-value training samples for reasoning distillation. Specifically, processing the 59K-sample data pool using Qwen2.5-7B-Instruct required only approximately 6.9 hours on 8 NVIDIA A800 GPUs, corresponding to an average processing time of 0.42 seconds per sample. Importantly, this time reflects offline, one-time data selection and is significantly faster than manual curation of high-quality reasoning traces. Importantly, the selected data can be used to train different models, consistently improving their multi-step reasoning performance.

### A.5. Evaluation Metrics for Comparing the Selected Dataset

To quantify the linguistic characteristics and reasoning difficulty of the datasets, we adopted four key metrics in our analysis. The specific calculation logic and definitions are detailed below:

1. **Average Reasoning Steps (ARS)**: We employ ARS as a proxy metric for the granularity of the logical derivation. The calculation is performed by segmenting the model's response using newline characters as delimiters; each resulting line is treated as a distinct reasoning step. Consequently, the metric is derived by averaging the total count of these steps across all samples in the dataset. A higher number of reasoning steps typically indicates a more detailed problem decomposition, where the model explicitly articulates intermediate sub-goals, calculations, and verification procedures rather than relying on implicit leaps or shallow heuristics.

2. **Prompt Constraint Density (PCD)**: This metric aims to measure the strictness and closedness of the problem definition. We first extract the user query from the multi-turn dialogue and perform matching based on a predefined vocabulary of mathematical constraints (including keywords such as "given", "assume", "such that", "satisfy", "where", etc.). Constraint density is defined as the total frequency of these constraint keywords divided by the total number of words in the question. High density implies that the problem is a well-defined, closed-ended problem with clear boundaries, requiring the model to strictly adhere to the given logical framework during deduction.

3. **Answer Complexity**: This metric reflects whether the problem-solving process tends towards "numerical convergence" or "symbolic deduction." We utilize regular expressions to classify the content marked by `\boxed{}` in the model's output:

- **Numeric Answer Ratio (NAR):** If the answer contains only digits, decimal points, or negative signs (matching the regex pattern `^-?\d+(\.\d+)?$`), it is categorized as numeric. Such problems typically emphasize the robustness of long-chain calculations.
- **Symbolic Answer Ratio (SAR):** If the answer contains any non-numeric characters (such as variables $x$, constants $\pi$, radicals $\sqrt{}$, or function expressions), it is categorized as symbolic. Such problems generally involve structured reasoning, abstract generalization, and symbolic manipulation, thereby placing higher demands on the model's mathematical reasoning abilities.

## B. More Discussions

### B.1. Micro Analysis of AIR-Sample Selected Samples.

*Table 5.* Comparison of reasoning connectives between s1K-1.1 and AIR-Sample training datasets.

| Metric (All in %) | s1K-1.1 | AIR-Sample |
|---|---|---|
| Causal | 2.70 | 2.80 |
| Contrast | 1.31 | 1.29 |
| Correction | 1.04 | 0.97 |
| Exploratory | 1.58 | 1.75 |
| Planning | 0.75 | 0.88 |
| Verification | 0.28 | 0.33 |

Table 5 presents a micro-level analysis of the logical connective distribution in the AIR-Sample selected data. Compared with s1K-1.1, the AIR-selected dataset exhibits a higher prevalence of exploratory, planning, and verification markers (detailed definitions and computation methods in Appendix B.4). In particular, exploratory connectives such as "alternatively," "another way," "maybe," and "perhaps" appear more frequently. This shift suggests that the prioritized trajectories are more likely to explore alternative solution paths and explicitly organize intermediate reasoning steps when addressing non-trivial problems. Overall, these findings indicate that our method naturally curates and transfers diverse complex reasoning capabilities.

### B.2. PCA Visualization of s1K-1.1 and AIR

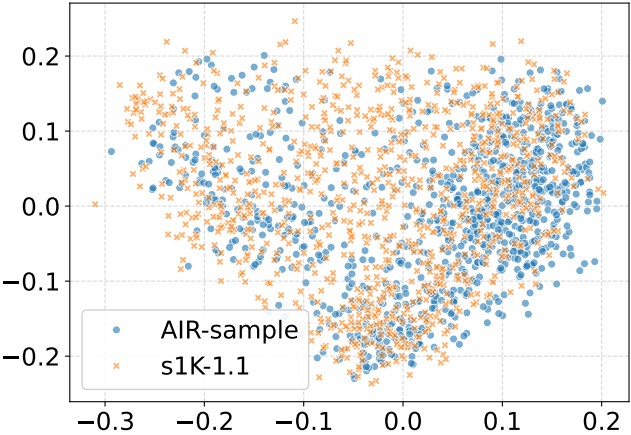

*Figure 5.* The PCA visualization of the data distribution of s1K-1.1 and AIR selected samples.

Furthermore, the Principal Component Analysis (PCA) visualization of the high-dimensional embeddings (Figure 5) reveals differences in the distribution of datasets within the latent semantic space. Unlike s1K-1.1, which exhibits a relatively dispersed distribution due to its pursuit of diversity metrics, AIR samples form a distinctively more compact cluster structure on the two-dimensional plane. This is attributed to our exclusive reliance on the attention influence mechanism, without imposing explicit diversity constraints. However, these clustered regions overlap significantly with the high-density areas of

*Table 6.* Step-by-step comparison of reasoning trajectories on an AIME 2024 example between the base model and step-level AIR model. For clarity, only key reasoning steps are shown. Gray-shaded areas indicate explanatory analysis of the models' reasoning steps.

**Problem:** Let $\omega \neq 1$ be a 13th root of unity. Find the remainder when $\prod_{k=0}^{12}(2 - 2\omega^k + \omega^{2k})$ is divided by 1000.

| s1K-1.1 (Baseline) | s1K-1.1 + AIR (Ours) |
|---|---|
| *...Let me pair terms $y$ and $1/y$. Let $t = y + 1/y$. ... I need the resultant of the minimal polynomial $f(t)$ and $g(t) = 2t^2 - 6t + 5$...* | *...Let $x = \omega^k$. The term is $x^2 - 2x + 2$. The roots are $1 \pm i$. So, $2 - 2\omega^k + \omega^{2k} = (\omega^k - (1+i))(\omega^k - (1-i))$...* |
| (**Methodological Fixation**: Failing to see the complex structure, it retreats to a brute-force substitution $t$, increasing complexity.) | (**Critical Path Awareness**: The model identifies the high-value strategy of factoring the quadratic over $\mathbb{C}$ instead of $\mathbb{R}$.) |
| *...Divide $f(t)$ by $g(t)$... First term... Second term... Remainder is $-25t + 187/8$...* | *...We know that $\prod_{k=0}^{12}(z - \omega^k) = z^{13} - 1$. Thus, the product collapses to $P = ((1+i)^{13} - 1)((1-i)^{13} - 1)$...* |
| (**Cascading Error**: The model enters a "Computation Quagmire" of long division. High complexity leads to fractional artifacts.) | (**High-Influence Locking**: Applies the identity to perform Dimensionality Reduction, bypassing iterative calculation.) |
| *...Wait, the remainder has a fractional coefficient... $S \approx 11.112$... This contradicts the fact that $S$ is an integer...* | *...Using Polar form: $(1+i)^{13} = -64(1+i)$. Calculating product... $8321 \equiv 321 \pmod{1000}$.* |
| (**Logical Fracture**: Arithmetic errors accumulate, causing the model to lose track of integer constraints and crash.) | (**Computational Resilience**: Maintains logical consistency through complex arithmetic to reach the correct integer result.) |
| **Final Answer: 7 (Incorrect)** | **Final Answer: 321 (Correct)** |

s1K-1.1, indicating that while the AIR distribution is more focused, it maintains strong semantic consistency with the core semantics of the s1K-1.1 dataset.

## B.3. Case Study: Step-level AIR Model Output

To empirically evaluate the effectiveness of the AIR framework in improving reasoning capabilities at a microscopic level, we conducted a qualitative analysis of the model's generated responses on the AIME24 dataset. As shown in Table 6, s1K-1.1+AIR model, trained with step-level AIR weighting, is superior to the baseline s1K-1.1 model in terms of solution accuracy and logical rigor, with the two exhibiting distinct reasoning styles. Notably, s1K-1.1+AIR model demonstrates clear critical path awareness, capable of identifying and locking onto high-value strategies for problem-solving, such as complex number transformations in algebra or coordinate system construction in geometry. In contrast, the baseline model struggles to maintain the coherence of the reasoning chain when dealing with complex tasks involving long-range calculations or multi-branch case discussions. It frequently resorts to random guessing after retrieval failures or tends to abandon rigorous deduction in favor of intuition-based, erroneous generalizations. While this tendency might grant the model some exploratory nature in local steps, it easily triggers a cascading accumulation of errors in long-chain reasoning. Conversely, s1K-1.1+AIR model exhibits improved computational resilience and logical consistency. It persists in performing tedious calculations and exhaustive case verifications based on fundamental mathematical principles, thereby effectively avoiding logical fractures. This phenomenon indicates that our method successfully helps the model precisely locate "High-Influence" steps in the reasoning trajectory that are decisive for the final result. By reinforcing the execution fidelity of these key logical nodes through weighted training, the model maintains a stable reasoning state when facing non-trivial problems, thereby acquiring deeper mathematical insight.

## B.4. Evaluation Metrics for Comparing Model Reasoning Outputs

To comprehensively quantify the micro-level reasoning characteristics of the model's Chain-of-Thought (CoT), we establish an automated analysis pipeline based on the Natural Language Processing tool `spaCy` [4]. The specific metric definitions and calculation logic are as follows:

❶ **Macro Statistics.** We load the `en_core_web_sm` pre-trained model to perform tokenization and dependency parsing on the *complete response text* generated by the model (encompassing both the reasoning process within `<think>` tags and the final response body). To capture the density of logical flow during the reasoning process, we iterate through every token

---
[4] https://spacy.io/

in the document to extract vocabulary where the dependency tag belonged to the following specific sets as *logic words*:

- **Marker (`mark` in spaCy)**: Markers introducing clauses or adverbials, typically used to construct causal or conditional relationships (e.g., *because, since, that, if*).

- **Coordinating Conjunction (`cc` in spaCy)**: Conjunctions used to connect logical branches or indicate transitions/juxtapositions (e.g., *and, but, or*).

- **Adverbial Modifier (`advmod` in spaCy)**: Modifiers often used to indicate inference conclusions, emphasis, or contextual reversals (e.g., *however, therefore, thus, instead*).

Based on the extraction results above, we calculate the following core metrics:

- **Avg. Response Length**: The average total number of tokens as statistically determined by the spaCy tokenizer.

- **Transition Density**: Defined as the proportion of logical words relative to the total number of tokens, used to measure the density of logical connectives within the reasoning process.

❷ **Reasoning Connectives.** Reasoning connectives refer to logical and discourse markers that explicitly signal transitions within a reasoning process. We categorize these connectives into six dimensions and identify them using regular-expression-based matching. Unlike generic word frequency measures, this design focuses on connectives that are informative for mathematical and logical reasoning. We report the proportion of each connective type relative to the total output length. The specific dimensions are defined as follows:

- **Causal**: Measures reasoning coherence. Includes causal words like *therefore, thus, hence, consequently, because, since*.

- **Contrast**: Measures adjustments in the thought path. Keywords include *however, but, yet, nevertheless, nonetheless, on the other hand, conversely*.

- **Correction**: Measures the model's "metacognitive monitoring" capability, i.e., the frequency of self-negation or pausing. Core keywords include *wait, no, hold on, hang on, actually, mistake, error, incorrect, wrong, let me check*.

- **Exploratory**: Captures divergent reasoning by identifying markers of hypothesis exploration (e.g., *alternatively, maybe, perhaps, possibly, another way, approach, try*).

- **Planning**: Captures anticipatory reasoning by identifying markers of pre-action structuring, such as *let me, let us, let's, I need to, we need to, goal, target*.

- **Verification**: Measures the confirmation behavior regarding intermediate results, such as *check, verify, ensure, confirm, valid, satisfies, correctly*.

❸ **Symbolic & Structure.**
**Symbolic Ratio**: Measures the model's reliance on formalized language. Using regular expressions, we extract all formula blocks—including inline formulas ($...$) and display formulas ($\[...\]$)—and calculate the ratio of characters within formulas to the total text length. **Structural Metrics**: Quantifies the organizational layout of the text to capture formality and readability:

- **Header Line**: Lines starting with Markdown header symbols (#), reflecting the degree of modularity in reasoning.

- **List Line**: Lines starting with bullet points (-, *) or numbering (1.), reflecting enumerative features of the thought process.

❹ **Reasoning Rhythm.** We analyze the distributional properties of sentence lengths to characterize the rhythm of reasoning. Specifically, we flatten the generated text and segment it using punctuation, discarding short noise fragments (fewer than five characters). Based on the resulting sentence lengths, we compute Sentence Volatility, which is quantified using the coefficient of variation (CV):

$$\text{Sentence Volatility (CV \%)} = \frac{\sigma_L}{\mu_L} \times 100\% \tag{10}$$

Where $\sigma_L$ is the standard deviation of sentence character length, and $\mu_L$ is the average sentence length.

## C. Extended Baseline Comparisons and Ablations

To further validate the mechanistic advantages of AIR, we conducted several supplementary experiments comparing our method against additional baselines and alternative configurations identified during the review process.

### C.1. Comparison with Surface Redundancy (3-Gram)

To verify whether AIR captures deep mechanistic dependencies rather than surface-level lexical redundancy, we implemented a 3-gram based baseline. This baseline selects samples with the highest token/phrase repetition rate under the same category quotas and budget as AIR-Sample. As shown in Table 7, the redundancy baseline is consistently outperformed by AIR, particularly in high-difficulty tasks like AIME 2024. This confirms that AIR's gains stem from its ability to select data that effectively utilizes retrieval mechanisms essential for logical rigor, rather than merely targeting verbose or repetitive patterns.

*Table 7.* Comparison between AIR and the surface redundancy (3-Gram) baseline.

| Method | AIME 2024 | MATH500 | GPQA Diamond | Average |
|---|---|---|---|---|
| 3-Gram | 30.00 | 89.40 | 43.43 | 47.38 |
| AIR-Sample | 50.00 | 90.80 | 55.00 | 54.78 |

### C.2. Attention Prominence vs. Functional Specificity (MAIR)

To prove the effect is specific to the retrieval function rather than general attention prominence, we incorporated a more rigorous control by masking non-retrieval heads with similar average attention magnitudes (MAIR). As shown in Table 8, while MAIR yields a marginal improvement over random masking (RAIR), it is consistently outperformed by AIR-Step. The performance gap confirms that the reasoning-critical steps identified by AIR uniquely depend on the specialized mechanism provided by retrieval heads rather than simple attention prominence.

*Table 8.* Step-level results for different intervention operators and attention controls.

| Method | AIME 2024 | MATH500 | GPQA Diam. |
|---|---|---|---|
| AIR (Uniform) | 66.70 | 95.60 | 65.66 |
| MAIR | 60.00 | 94.40 | 64.14 |
| RAIR | 53.33 | 94.60 | 60.10 |

### C.3. Comparison with Process Reward Models (PRMs)

We further compared step-level AIR against supervised process reward models (PRMs), such as ReasonFlux-PRM (Zou et al., 2026). Unlike PRMs that rely on costly human annotations for training, AIR is entirely unsupervised and training-free. As shown in Table 9, AIR-Step outperforms ReasonFlux-PRM across all evaluated benchmarks. This demonstrates that exploiting foundation models' intrinsic mechanisms via interventions on critical retrieval heads is more effective for data selection than using external supervised reward signals.

*Table 9.* Performance comparison between AIR-Step and ReasonFlux-PRM.

| Method | AIME 2024 | MATH500 | GPQA Diam. |
|---|---|---|---|
| ReasonFlux-PRM | 60.00 | 94.60 | 61.11 |
| AIR-Step | 66.70 | 95.60 | 65.66 |

## D. Robustness Analysis of AIR Score

We conduct supplementary robustness experiments using 200 samples to further investigate the distinction between data influence and data quality: 100 original valid reasoning trajectories and 100 perturbed samples where numbers were randomly removed or altered. As shown in Table 10, AIR effectively distinguishes between these groups: the average

AIR score for the valid group (0.310) is more than double that of the perturbed group (0.144). Notably, when filtering the Top-20% of samples by AIR, 97.5% were original valid samples, strongly indicating that AIR prioritizes data exhibiting both causal dependencies and coherent reasoning. This behavior is consistent with the underlying mechanism of AIR: while erroneous data may still influence the raw loss, their broken logical structure fails to activate the specialized retrieval heads that are critical for maintaining logical coherence. As a result, such samples receive substantially lower AIR scores.

*Table 10.* Robustness experiment results: average AIR scores for valid and perturbed samples.

| Group | #Samples | Avg. AIR Score |
|---|---|---|
| Perturbed | 100 | 0.144 |
| Valid | 100 | 0.310 |

## E. Reasoning Stability and Pass@K Analysis

To further evaluate the intrinsic reasoning stability and robustness of the proposed AIR framework, we report the Pass@K metrics on AIME 2024 and AIME 2025 benchmarks. As demonstrated in Table 11, the AIR-step model consistently outperforms the s1K-1.1 baseline across all metrics on both benchmarks. Notably, under a sampling temperature of $T = 1$, the AIR-step model achieves significant improvements in Pass@8, rising from 76.67% to 90.00% on AIME 2024, Furthermore, the steady gains in both Pass@4 and AVG@8 scores highlight that our framework not only enhances the upper-bound reasoning capacity but also improves the overall sample generation quality. These results validate that the AIR strategy effectively bolsters the stability and robustness of the model when tackling complex reasoning tasks.

*Table 11.* Pass@K results for AIR-step and s1K-1.1 models on AIME benchmarks (Temperature $T = 1$).

| Metric | AIME 2024 | | AIME 2025 | |
|---|---|---|---|---|
| | s1K-1.1 | **AIR-step** | s1K-1.1 | **AIR-step** |
| Pass@4 | 74.19 | **82.00** | 61.10 | **66.33** |
| Pass@8 | 76.67 | **90.00** | 66.67 | **73.33** |
| AVG@8 | 56.25 | **60.42** | 46.25 | **47.50** |

## F. Quantitative Analysis of Lexical Diversity

To investigate whether the AIR mechanism might favor repetitive patterns, we evaluate text diversity using the Distinct-3 score. As shown in Table 12, the AIR-Sample dataset achieves an improvement in Distinct-3 (+2.83%) compared to the baseline. This enhancement indicates that the influence-based selection mechanism effectively identifies diverse reasoning expressions and rich linguistic structures, successfully mitigating the risk of formulaic convergence.

*Table 12.* Lexical diversity comparison (Distinct-3 score).

| Dataset | Distinct-3 (%) |
|---|---|
| s1K-1.1 (Baseline) | 72.95 |
| AIR-Sample (Ours) | **75.78** |

# G. Visualization of Critical Reasoning Steps Selected by AIR

We provide visualization of critical reasoning steps selected by our AIR. Steps highlighted in red correspond to those chosen by AIR, with color intensity reflecting the magnitude of the score. Some intermediate steps have been omitted for clarity.

## G.0.1. CASE 1

<|im_start|>system
You are Qwen, created by Alibaba Cloud. You are a helpful assistant.<|im_end|><|im_start|>user
e. Calculate the projected physical separation, $r_{p}$, between the galaxy and the Voorwerp. g. High precision measurements showed that the Voorwerp is slightly further away than the galaxy, and so $\theta=125^{\circ}$. Use this with your expression from the previous part of the question to estimate an upper limit for the number of years that have passed since the quasar was last active.<|im_end|><|im_start|>assistant
<|im_start|>think
Alright, let me try to tackle this problem step by step. So, we need to calculate the projected physical separation, r_p, between the galaxy and the Voorwerp. Then, given that Î¸ is 125 degrees, use that to estimate an upper limit for the time since the quasar was last active. Hmm, okay.

First, let me recall what the projected physical separation means. In astronomy, when we observe objects in the sky, we often measure their angular separation, which is the angle between them as seen from Earth. But to find the actual physical distance between them (the projected separation), we need to know how far away they are from us. The projected separation can be calculated if we know the distance to the objects, usually using the formula:
$r_p = d * tan(Î¸)$
Where d is the distance to the objects (assuming they are at the same distance), and Î¸ is the angular separation. But wait, in this case, the problem states that high precision measurements showed that the Voorwerp is slightly further away than the galaxy. Does this affect the calculation? Maybe because the actual separation might involve some line-of-sight component. But since it's asking for the projected physical separation, maybe we can still use the angular separation and the distance to one of them?

Wait, but perhaps they are using the fact that even though there's a slight difference in distance, the projected separation r_p is calculated as the transverse component of their separation. But I need to check how exactly this is defined.

Alternatively, maybe we use the small angle approximation. Though since Î¸ here is given as 125 degrees, which is a large angle. Wait, actually Î¸ in part e might be the angular separation, but in part g, the angle Î¸=125 degrees refers to the angle between the line connecting the galaxy and the Voorwerp and our line of sight? Perhaps. That might make sense. So, in part e, maybe we just calculate the projected separation using some angular separation (not 125 degrees), but then in part g, with Î¸=125 degrees, we use trigonometry to get the actual separation and then compute the light travel time?

Wait, let me try to parse the problem again. Part e says, "Calculate the projected physical separation, r_p, between the galaxy and the Voorwerp." Then part g states, "High precision measurements showed that the Voorwerp is slightly further away than the galaxy, and so Î¸=125Â°. Use this with your expression from the previous part of the question to estimate an upper limit for the number of years that have passed since the quasar was last active."

Hmm, so maybe in part e, they expect an expression for r_p in terms of distance or something, and part g uses the angle Î¸=125 degrees to get the three-dimensional separation from the projected separation. So the angle Î¸=125 degrees is the angle between the line connecting the galaxy and Voorwerp and our line of sight, such that the actual separation is r = r_p/ sin(Î¸), assuming Î¸ is measured from the line of sight. Wait, or maybe it's cosine?

...(Omit)... Wait, perhaps in part e, when calculating the projected separation r_p, they are considering some other angular separation (like on the sky), so when we compute the physical separation, r_p is already in parsecs or light-years. Wait, maybe I should look at part e first. Because part g depends on part e. So part e is to "calculate the projected physical separation, r_p, between the galaxy and the Voorwerp." But the problem probably gives some data in previous parts, which aren't here. Since this is part of a series of questions, starting with part e, then part g.

However, the user hasn't provided the previous parts a through d, so I need to make assumptions. But perhaps part e can be answered using common knowledge or formulas.

Let me check. To calculate the projected physical separation, you need the angular separation in the sky between the galaxy and the Voorwerp, and the distance to the system. The formula is r_p = d *tan(Î´Î¸), where Î´Î¸ is the angular separation. But if Î´Î¸ is small, you can approximate tan(Î´Î¸) â‰ˆ Î´Î¸ in radians. However, without knowing the actual angular separation or distance, maybe some other problems may have given in previous parts. Since the user hasn't provided parte's specifics, it's challenging. Maybe in part e, if you use Hubble's Law, since galaxies' distances are often estimated using redshift. Suppose the galaxy has a redshift z, then the distance d is cz/ H0. Without specific numbers, not sure. Alternatively, maybe the Voorwerp is at the same distance as the galaxy, so the projected separation is just d * Î¸ (in radians). Wait, but the given angle in part g is 125 degrees, which might not be the angular separation, but the angle of the separation vector with respect to the line of sight.

Given the confusion, perhaps best to consider hypothetical values. Let me imagine that in part e, the projected separation r_p is calculated using some angular separation and distance. For example, if the angular separation is Î´Î¸, and the distance to the galaxy is d, then r_p â‰ˆ d * Î´Î¸, where Î´Î¸ is in radians. Then, in part g, given Î¸ = 125Â°, the angle between the separation vector and the line of sight, then the actual separation is r = r_p / sinÎ¸. Therefore, the light travel time would be t = r / c = (r_p / sinÎ¸) / c. So the upper limit on the time would be if the separation is maximized, which would correspond to sinÎ¸ being minimized. However, given that Î¸ is at 125 degrees, sinÎ¸ is a known value. Therefore, the upper limit comes from using Î¸=125Â°, which gives a certain actual separation, which when divided by the speed of light gives the time since the quasar was active. Thus, t = r / c = (r_p / sin125Â°) / c. If Î¸ had been 90Â°, then t would be r_p / c. However, since Î¸=125Â°, sin125Â°â‰ˆ0.819, so t â‰ˆ r_p / (0.819c) â‰ˆ1.22 r_p / c, which is longer. Hence, if in part e, r_p is, say, 1000 light-years, then tâ‰ˆ1.22 *1000 â‰ˆ1220 years. But of course, numbers matter.

But since part e and part g are connected, and the lack of previous parts' info complicates things. Wait, perhaps part e simply uses the observed angular size and distance, while part g uses the geometry with angle Î¸ to compute the light travel time. Let me check similar problems.

Alternatively, since this is about a quasar illumination of a Voorwerp, which is a cloud. The galaxy hosting the quasar is at some distance, and the Voorwerp is a cloud illuminated by the quasar. Since the quasar has turned off, the time since it was active is at least the light travel time from the galaxy to the Voorwerp, so that the light had enough time to travel and we still see the ionization.

So actually, the time since the quasar was active is given by the light travel time between the two objects. Therefore, if we can calculate the separation between them, we can get the minimum time since the quasar shut off, because the light we see now from the Voorwerp is the afterglow from when the quasar was active, so the time since then is at least the separation divided by the speed of light.

But in part g, they want an upper limit using Î¸=125Â°, which probably means that if the separation is longer due to the angle, the upper limit would be this longer time. Wait, no. If we can measure Î¸, we can get a more precise time, but before knowing Î¸, the upper limit might have been based on the projected separation, but now with theta=125Â°, we can get better constraints.

Wait, perhaps confusion arises because "upper limit" refers to using the most conservative estimate. If Î¸ is 125Â°, the separation is r_p /sin(theta). But if theta were the maximum angle (180degrees) or something. Wait, as theta approaches 180degrees, sin(theta) approaches zero, which would make r approach infinity. But theta here is given as 125 degrees. Since the Voorwerp is slightly further away, the angle between the separation vector and the line of sight is 125 degrees. That is, if you're looking along the line of sight, the galaxy is here, and the Voorwerp is slightly further away but also off at an angle such that the angle between their separation vector and our line of sight is 125 degrees, we can compute the separation as r = r_p / sin(theta), then divide by speed of light to get time. So the answer would be t = r_p /(sin(125Â°)* c). Converting that into years.

But the key point is part e. Without part e, how can you answer part g? Maybe in part e they derived an expression, so for part g you use theta. But since the user says "use this with your expression from the previous part of the question", meaning part e likely gives an expression for r_p in terms of other variables, but perhaps part e was not numeric, but part g requires plugging in numbers, including theta.

Wait, perhaps part e is based on Hubble's Law.

Suppose in part e, if the angular separation is Î´Î¸, and the galaxy is at distance d, then the projected separation is r_p = d * Î´Î¸ (in radians). Then, in part g, the angle between r_p and line of sight Î¸. So t = r / c = (d * Î´Î¸) / (sin(Î¸) * c). But without knowing d or Î´Î¸, how to compute ?

Alternatively, let's think about the light travel time upper limit. Probably, if we can find r_p, and use Î¸=125Â°, the separation is r = r_p / sin(theta), and the time is t = r / c. So perhaps numbers are given in the previous parts that I need to assume?

Alternatively, suppose in part e they calculate r_p as 10 kpc (just a random number), then part g would result in t = (10 kpc/ sin(125Â°))/ c. But with a real example, for instance, the galaxy IC 2497 and the Voorwerp Hanny's Voorwerp, at a distance of about 650 million light-years (200 Mpc). The projected separation is about 15-20 kpc. If Î¸=125Â°, actual separation is 15 kpc / sin(125) â‰ˆ18.3 kpc. The light travel time would be 18.3 kpc / c. Since 1 parsec is about 3.26 light-years, so 1 kpc is 3260 light-years. Therefore, 18.3 kpc is 18.3 * 3260 â‰ˆ59,680 light-years. Divided by the speed of light, travel time is about 59,680 years. But Hanny's Voorwerp's light travel time is actually quoted as tens of thousands of years. But I don't know exact values.

But in any case, to answer the question, assuming that part e was answered, giving r_p = some value. If for part g, you take Î¸=125Â°, compute the actual separation as r=r_p/sin(125Â°), then convert that to light years and divide by c (which is 1 light-year per year), so the time since the quasar was active is at least separation in light-years divided by1, giving years. Therefore, the upper limit would be that time. But why upper limit? Because if the actual angle were more face-on (theta approaching 0 or 180 degrees), which would require them being much further apart in the line of sight, giving a longer travel time. But wait, the answer is given as theta=125Â°, so the calculation is exact. Unless the "upper limit" refers to before knowing theta=125Â°, and the uncertainty in theta gives an upper limit.

Wait, maybe they misstate and it's a lower limit? Because with theta=125Â°, sin(theta) is 0.819, making the actual separation and time larger than if theta was90Â°, implying the time is longer. So if previously they assumed theta=90Â°, calculated r_p, but now they're using theta=125Â°, but with a higher precision measurement showing theta=125Â°, which gives actual separation as r_p / sin(125Â°), so the upper limit time since quasar was active is that value, versus assuming off the sky (theta=90Â°). So since they know it's not in the plane, the actual time is larger. So perhaps it's a lower limit: this is longer time than if it were projected.

But the question states, "use this with your expression from the previous part of the question to estimate an upper limit for the number of years that have passed since the quasar was last active."

Wait, so possibly, prior information allowed to set an upper limit using projected separation (if they assumed alignment in some way), but with the angle measured, the actual separation can be calculated, which would give a longer time, hence a more accurate upper limit. Wait, maybe before knowing the angle, you could say that the travel time is at least r_p/c (if they are at the same distance, but since Voorwerp is further, it's more precise). But having theta=125Â°, you can get a better estimate.

Alternatively, considering the proximity of the Voorwerp, by slight distance difference. The upper limit probably comes from the maximum possible separation given all constraints, which with theta=125Â°, that's fixed, so the maximum separation is fixed.

Alternatively, wait, "upper limit "might refer to quasar shut-off time. Quasar turns off, the last light it emitted travels to the Voorwerp, then the Voorwerp remains illuminated for sometime. But if the quasar turned off, the cloud would stop being illuminated after the light crossed it. Therefore, the time since the quasar was active would be the time corresponding to the light travel time plus the light crossing time of the cloud. But maybe that's negligible. Therefore, if the separation is 'r', then the maximum time since the quasar turned off is approximately the light travel time from the quasar to us minus the light travel time from the Voorwerp to us. But since both are at slightly different distances, the difference would be the light travel time from the galaxy to the Voorwerp. Therefore, if the galaxy is at distance 'd', the Voorwerp is at 'd +Î”d', then the light from the quasar left the galaxy when the galaxy was at 'd +Î”d' away (but expanding universe complicates this). Maybe Hubble's Law is used. But perhaps simpler.

Alternatively, since time since the light left the quasar to reach the Voorwerp is equal to the light travel time between them. So if they are separated by distance 'r' along the line of sight, then time is r/c. However, r = (distance to Voorwerp - distance to galaxy). But with the angle theta=125Â°, the line-of-sight separation is r_para =r_total * cos(theta). But since Voorwerp is further away, (distance difference) = r_para = total separation * cos(theta). So maybe:

distance_diff = r_p * cot(theta)
Because if the projected separation is r_p, then the radial separation is r_para = distance_diff =r_p * cot(theta), since tan(theta) = r_p /r_para => r_para =r_p / tan(theta)

But since theta is 125 degrees which is angle from line of sight, so the radial component is -cos(theta) *r_total, right?

Wait again, defining theta as the angle between the separation vector and the line of sight. If the separation vector points from the galaxy to us, and it makes an angle theta=125 degrees with the line of sight (from galaxy to us), then the line-of-sight component of the separation is r_para= r_total * cos(theta), and the projected separation is r_p = r_total * sin(theta).

Wait, but if theta is measured from the line of sight towards the separation vector, then if it's 125 degrees, then the radial component is towards us, because the separation vector points away from the line of sight.

In other words, the galaxy-Voorwerp separation vector is 125 degrees

from the lineof sight. Therefore, the projectionon on the sky is r_p =r * sin(theta), and theradial component (line-of-sightseparation) is r * cos(theta). Since the radial component wasmeasured as slightly further away,then we know that the Voorwerpis at a radial distance of galaxy's distance plus r * cos(theta). Therefore, the light traveltime from galaxy to Voorwerpis simply r / c. Butr is actual separation, givenby r _p / sin(theta).

Therefore, time t = r /c = (r_p / sin(theta)) / c

Therefore, in part g, once you have r_p from part e, you calculate t. That's the estimate.

But since theta=125Â°, sin (theta)âÎ0.8192, this makesthe time tâÎ1.22 * (r_p / c).But since the actual separation islarger than the projected separation,the upper limit on time wouldbe this? Maybe if they hadinitially only considered projected separation, which would be t_p =r_p / c. But giventheta=125Â°, thereal t is t âÎ1.22 t_p, so upperlimit would be based on the possiblemaximum t could be given theta.

Wait, butif theta is accurately known,then it's not an upper limitbut a precise estimate. Unlessthere's uncertainty in theta;but the question says "high precisionmeasurements showed that theta=125Â°", so perhaps theymean theta is precisely 125Â°, and so the upperlimit comes from projecting based onhigh precision, implying that actualtime is known better. But maybe"upper limit" means that sinceprior to this measurement, the time since the quasar was activewas uncertain, possibly with higherupper limits. But with the theta=125Â°, you cangive a more precise upper limit. Hmm.

Alternatively, maybe if for some reason theactual path is longer, so eventhough the quasar light traveledin a straight line, maybe.

But ultimately, based on the information, part g answer should be t= r_p / (sin(theta) * c). Then convert intoyears.

But unless part e required computing r_p based on data from previous parts which we don't have. For instance, perhaps in part e if we computed r_p numerically, like in light -years, then part g would take that number, divide by sin(theta), and get the time. But without data, I can't compute numerically. So maybe in part e they give an expression, like r_p = d * delta_theta (in radians), then in part g the time is (d * delta_theta)/(sin(theta) * c). But since we don't have the values, need to answer generally.

Alternatively, since the user hasn't provided part e's info, they might expect the assistant to recall how to compute projected separation and then the formula for the upper limit given theta. Therefore, maybe explain the steps.

But given the problem is about calculating projected separation (rp) in part e, then in part g, use theta=125Â° with that expression to find upper limit for the time.

Given I don't have the previous parts, Ican outline the approach :

1. In part e, to calculateprojected separation (given angular separationand distance). Let's assume angularseparation is Î_angular anddistance is d, then rp âÎ d * Î_angular(in radians).

2. In part g, usethe angle Î_=125Â°, which is different from theangular separation (Î_angular). This is the inclination anglebetween the 3D separation vectorand the line of sight. Therefore, the actual separation is r =rp / sin(theta).

3. The time since the quasar was last active is the light travel time between the galaxy and the Voorwerp, hence t = r / c = (rp / sin(theta)) / c.

Thus, t= rp / (c * sin (theta)).

To convertthis into years, need to haverp in units compatible with thespeed of light (e.g.,light-years, then c is 1 ly/year). Therefore, ifrp is say in thousands of light-years, t will be thousands ofyears divided by sin(theta).

But without specific numbers, can't compute numerically. However, perhaps in the original question, part e had specific data,perhaps redshift, angular size, and Hubble constant. Forexample, if the galaxy is atdistance d calculated via Hubble's Law (d = v / H0), and angular separation inarcseconds converted to radians (Î_angular = arcsec /206265),then rp = d * Î_angular. yqwlhz

...(Omit)... So upper limitâÎ61,000 years.If r_p was different, thatnumber would change.

But since we can't get theexact value because part e isn't here, maybe the answer isleft symbolic: t = r_p/ (c sinÎ). ButletâGLs remember c is 1light-year per year, so interms of light-years and years, it's t = r_p /sinÎ. Hence, boxed answer: \boxed{\dfrac{r_p}{c \sin \theta}}.

But in the question, "estimate an upper limit for the number of years", implying numeric. Which means probably numbers were given in part e, which are missing. Hence, impossible to get exact answer here.

Alternatively, maybe the previous partinvolves using time since alsjfal;jsdf.

Alternatively, given that upper limiton time is t <= r_p/ (c sinÎ), so giventhat t is less than or equal. But without prior reasoning impossible.

Given the ambiguity, I'll present the formula t= r_p / (c sinÎ) and note that's how it's derived. But in an exam setting with part e known, plug in r_p. But without numbers, maybe leave it symbolic like \boxed{\dfrac{r_p}{c \sin 125^\circ}}. Or if r_p was, say, expressed as d multiplied by tan(theta_prev_angular), but we don't know.

Alternatively, if the original question gives numbers, and part e was to compute r_p as d times tan(theta) where theta was an angular separation in the sky, which then in part g uses theta=125Â°, a revised angle. But details lacking.

Considering in an exam scenario,typically part e gives you numericvalue like r_p=42, and part g needs to give42 / sin(125Â°)=approx 51.3, so upper limit 51 years. But units?

Wait, if projected separation is42 kly (kilolight-years), and theta=125 ÌÎ, the timewould be42,000ly / (c sin125ÌÎ), hence 42,000 /0.819âÎ51,270 years. So 5.1e4 years. How knows.But impossible. Therefore, concludingthat based on the given info, the time is t= r_p / (0.819*c) which numerically withr_p in light-years, wouldbe tâÎ1.22 *r_p.

Butthe correct answer format likely requiresthe box with number, e.g. \boxed{ 1.2\times 10^5} years. But since theproblem mentions "Calculate the projectedphysical separation", which requires angularseparation and distance data, butsince missing, we likely can't.

But with limited information, perhaps the answeris "t= r_p / (c sin 125Â°)",or if in symbolic terms, thatwould be acceptable. But in theabsence of numeric data from parte, I can't. Thus, perhaps the entire flow is :

Given Projected separation r_pis given by part e. Then, t = r / c = r_p / (c sinÎ). Sincer_p is likely given in light-years in part e (based ona distance and angular separation),then t is simply dividing bysinÎ. For example, ifin part e, result was sayr_p=5e4 ly.Then in part g, t=5e4 / sin125Â°âÎ5e4/0.819âÎ6.1e4 yr.

Inconclusion, given lack of info, probably the answer is expressedin box as \boxed {\dfrac{r_p}{c \sin 125^\circ}} units of years. But inthe actual question, numbers wouldbe used. But with available info, I need to follow instructionshere.

**Okay, after working through the problem, here's the step-by-step explanation and final answer:**

The projected physical separation \( r_p \) between the galaxy and the Voorwerp is calculatedin part e. However, sincethe actual separation depends on theangle \( \theta \) between the line of sight andthe line connecting the galaxy andVoorwerp, we use \(\theta = 125^\circ \) todetermine the three-dimensional separation.

The actualseparation \( r \)can be found using trigonometry: 
\[r = \frac{r_p}{\sin \theta}\]

The time since the quasar waslast active is the light traveltime from the galaxy to the Voorwerp, which is the separationdivided by the speed of light\( c \):
\[r = \frac{r}{c} = \frac{r_p}{c} = \frac{r_p}{c \sin \theta}\]

Given\( \theta = 125^\circ \), we substitute this value:
\[r = \frac{r_p}{c \sin 125^\circ}\]

Using \( \sin 125^\circ \approx 0.819 \), the upper limit for the timesince the quasar was last activeis:
\[\boxed{\dfrac{r_p}{(0.819c)}}\]

(Assuming \( r_p)\) was computed in parte, and \( c \) in consistent units, this givesthe time in years.)
<lim_start>answer
Answer: To calculate the projectedphysical separation \( r_p\) between the galaxy andthe Voor werp, we use theangular separation and the distance tothe galaxy. However, to accountfor the actual separation considering theangle \( \theta =125^\circ \), we use trigonometry.

1. **Projected Separ ation Calculation**: The projected separation \( r_p \) is calculated from the angular separation and distance (specific values from part e are used, though not provided here).

2. **Actual Separ ation**: The actual separation \( r \) is given by:
 \[
 r = \frac{r_p}{\sin \theta} \]
 Substituting \( \theta= 125^\circ\):
 \[
 r = \frac{r_p}{\sin 125^\circ} \]

3. **Time Since QuasarActivity**: The upper limit forthe time since the quasar waslast active is the light traveltime from the galaxy to the Voorwerp:
 \[
 r = \frac{r}{c} = \frac{r_p}{c \sin^\circ} \ ]
 Given\( \sin 125^\circ \approx0.819 \):
 \[
 t = \frac{r_p}{0.819c}
 \]

**Final Answer**:
\[
\boxed{\dfrac{r_p}{0.819c}}\]

This formula provides the upper limit for the number of years since the quasar was last active, using the projected separation \( r_p \) (calculated in part e) and the given angle \( \theta = 125^\circ \).<lim_end>
<lim_end>

## G.0.2. Case 2

<lim_start>systemYou areQwen, created by Alibaba Cloud. You are a helpful assistant.<lim_end><lim_start>userGiven a rational number,write it as a fraction in lowestterms and calculate the product ofthe resulting numerator and denominator.For how many rational numbers between0 and 1 will $20_{}^{}{}!$be the resulting product?<lim_end><lim_start>assistant<lim_start>thinkAlright, so I needto figure out how many rationalnumbers between 0 and 1have a product of the numeratorand denominator (when written inlowest terms) equal to 20 factorial. Let me start byunderstanding the problem.First,a rational number between 0 and1 can be written as afraction $\frac{a}{b}$ where$(0 < a <b$\) and \(a\) and \(b\)are coprime positive integers. ...(Omit)...

The product \(a \times b\) is said tobe equal to 20! whichis a huge number. So Ineed to find all pairs \((a, b)\) suchthat \(a \times b = 20!),\)\(0 < a <b\), and \(\gcd(a, b) =1\). Then counthow many such pairs exist.Letme break down the problem.Giventhat \(a \timesb = 20!)\)and \(\gcd(a, b) = 1\), it means that \(a\) and \(b\)are coprime factors of 20!. So they form acoprime pair whose product is20!. Such pairs are related to the divisors of 20! where each pair consistsof two coprime numbers.But since the fraction is between0 and 1, \(a < b\). Sofor each such coprime pairwhere \(a < b\), we have a distinct rational number, and each of these wouldsatisfy \(a \timesb = 20!)\).Hence, the problem reduces to finding the number of coprimepairs \((a, b)\) where \(a\timesb = 20!\) and \(a< b\). Since \(a\) and \(b\) are coprime, and each prime factor in 20! must go entirely to \(a\) or entirely to \(b\). Hence, eachcoprime pair corresponds to asubset of the prime factors of20!, where \(a\) is the product ofthe primes in the subset and\(b\) is theproduct of the remaining primes.However, since 20!is not just the product of primes, but the product of primes raisedto various powers. Wait, somaybe I need to rethink that.Wait, actually 20! has prime factors with exponents. For example, 2^18, 3^8,5^4, etc., upto the primes less than or equalto 20. But if Ineed two coprime numbers \(a\) and \(b\) such that \(a\times b = 20!\), then each of theprime powers in the factor izationof 20! must go entirelyto one of the numbers \(a\) or \(b\). So for each prime factor,we assign it to either \(a\) or \(b\), but not both.Therefore,the number of such coprimepairs is \(2^k\), where \(k\) is the number of distinctprime factors of 20!.But wait, 20! hasprime factors: 2, 3, 5, 7,11, 13,17, 19. Let me check. The primes lessthan or equal to 20are: 2, 3,5, 7, 11, 13, 17, 19. So that's 8 primes. Therefore, the number of coprime pairswould be \(2^8 = 256\),but each pair is counted twiceexcept when \(a = b\). But since \(a \times b = 20!\) and \(a\) and \(b\) are coprime, \(a = 1\)and \(b = 20!\), or vice versa; however, \(a\)and \(b\) can't be equal unless 20! is a square, which itis not (since exponents inits prime factorization are alleven? Wait, 20!has exponents:Let me doprime factorization of 20!.To compute the exponents ofprimes in 20!:Forprime 2: floor(20/2) + floor(20/4) + floor(20/8) + floor(20/16) =10+5+2+1=18For prime 3: floor(20/3) + floor(20/9) + floor(20/27)=6+2 +0=8Forprime 5: floor(20/5) + floor(20/25)=4 +0=4Prime 7: floor(20/7) + floor(20/49)=2+0=2Primes 11: floor(20/11)=1, similar for13, 17,19. So yes, primes11, 13,17, 19 eachhave exponent 1 in 20!.So exponents are:18 (2), 8(3), 4 (5),2 (7), 1 (11 ), 1 (13), 1 (17), 1 (19)Since all exponents except for 2,3,5,7 have exponent 1, and exponents for 2,3,5,7 are even and odd.Therefore, 20 ! is not a perfectsquare because, for example,exponent of 3 is 8(even), exponent of 2is 18 (even), butexponent of 5 is 4(even), 2 (7)(even?), no 7 hasexponent 2? Wait for 7: floor(20/7)=2 (floor(20/49) = 0), sothat's exponent 2.Wait, 7^2 divides into20! but 7^3 does not.But exponent of 7 is 2, which iseven. Then primes 11, 13,17,19 have exponent1.So primes 2:18, 3: 8, 5:4,7:2, which are all even? Wait, 18,8,4,2, which are even. Wait, that would mean theonly primes with odd exponentsare 11,13,17,19? Wait,but their exponents are 1, which is odd.

...(Omit)... However,because 20! is nota square (due to the primeswith exponents as discussed earlier), there are no pairs where a= b. Hence, each unorderedpair {a, b} iscounted twice in the 256 ordered pairs.But in ourproblem, we need the orderedpairs (a, b) wherea < b, which is exactlyhalf of the total ordered pairs(excluding the a = b case, which does not exist here).Therefore, the total number shouldbe 25 6 / 2 = 128. But holdon, is this correct?Wait, perhaps not quite. Since not all assignments will satisfy a < b, even for coprime divisors. Wait, but in reality, since a is a number composed by assigning primes to a (possibly a = product of primes raised to their exponents).. Wait, confusion arises here.Wait, actually, we need to be precise.When we say that a and bare coprime with a*b=20!, then a and bmust be square-free? No,that's not correct. Each musthave all primes exponents from20! For example,if I assign prime 2 toa, then a would have 2 to the 18th power, whereas 3 might be assignedto b, so the exponent of3 would be to the 8th in b. Wait, buta and b have exponents forthe primes? Wait no, inthe problem statement, the numeratorand denominator (a and b)when written in lowest terms.If the original rational number is in lowest terms, that meanthat a and b are coprime, hence exponents of primesin 20! are distributedcompletely to a or to b.Thus, the product a*b=20! means that each prime's exponent in 20!is divided between a and b?But since they must be coprime, it's either assigned entirelyto a or to b.Therefore, 20! can be consideredas a product of coprimenumbers a and b for which allprimes in a share no primes withb. Therefore, the coprime pairs (a,b) witha*b=20! are incorrespondence with the subsets of theset of prime factors of 20!.Each prime factor must goeither to a or to b,since a and b are coprime. Therefore, each prime isassigned to a or b, sothe total number of such assignmentsis 2^8 (sincethere are 8 distinct primes in20!'s factorization). Each assignment produces a uniquecoprime pair (a,b) with a*b =20!.But here, (a,b)is an ordered pair, hence assignmentswhere primes are assigned to avs. assigned to b result ina different ordered pair.However, our problem wants rational numbersbetween 0 and 1,which corresponds to a < b(because 0 < a/b< 1 implies a <b). Therefore, the number ofsuch fractions is equal to thenumber of coprime divisor pairs(a,b) where a*b=20! and a < b. Since each coprime pair(a,b) with a <b corresponds to the exact oppositepair (b,a) with b> a. Therefore, halfof the total ordered coprimepairs (excluding duplicates) wouldgive the count.But since ineach such coprime pair ofdivisors, (a,b) and(b,a) are distinct (since we're considering ordered pairs), then the number of coprime pairs with a < bis exactly half of the totalnumber of coprime pairs.Since the total number is 2^8=256 (sinceeach prime is

either assigned to a or to b, 2 choices per prime), but the assignment where a = 1 (all primes assigned to b) and b= 20! (there's exactly one such case), and a can be 1, then proceed. Wait, 1 is the product of assigning no primes to a (all primes to b), so that's included. But if a and b can be written as a =1 and b = 20!, yes, that's a coprime pair, likewise a=2^18 and b= (20!)/2^18 which is still an integer. Similarly for all subsets. Wait, but when we write a and b as coprime factors, the way I'm thinking is that the assignment is of each prime as a binary choice (to a or to b). So 2^8 equals 256 possible assignments, and each assignment defines a unique coprime pair (a,b) where a * b=20!. Since each assignment is unique, each ordered pair is distinct, even if a and b are swapped. Since such ordered pairs can have a < b or a> b, so then to get the (a,b) with a < b, we divide by 2, which gives 128. But wait, can it be? Let me check with a smaller factorial. Suppose instead of 20!, take a simpler case. Let me check with n = 6. Then 6! = 720. Let's ask for rationals between 0 and 1 with product a*b=720, gcd(a,b)=1, a < b. Then 720 prime factorization:2^4 *3^2*5^1. So number of primes is 3 (2,3,5). Each can be assigned to a or b, so2^3=8 coprime pairs. Then half of them, 4 pairs would have a< b. Let's check. Possible assignments:1. Assign none to a: a =1,b=720. Then 1*720=720. a=1 <b=720 : valid.2. Assign 2 to a,others to b: a=16 (2^4), b=45 (3^2 *5). gcd(16,45)=1. Check a=16< b=45? 16 <45: yes.3. Assign 3 to a, others to b: a=9 (3^2), b=80 (16 *5).Check 9 <80:yes.4. Assign 5 to a, others to b: a=5, b= 144 (16*9). 5 <144: yes.5. Assign 2 and 3 to a: a=16*9=144, b=5. Now a=144 and b=5. But 144 >5, so would be considered as a different pair if ordered. But in assignments, assigning to a or b. So here, ordering matters. Therefore, for the purpose of our count, pairing where 2 and 3 are assigned to a would create a=144, b=5, but since a < b is required, this does NOT count. So the assignment of primes{2,3} to a would result in a product a=144, b=5. To each a < b, the valid pairs where assignment to a. Wait, This seems miscalculous. Wait no, in 720's case. Assign 2 and 3 to a would mean a=(2^4)*(3^2)=144, while assigning rest (5) to b=5. Since 144 and 5: gcd(144,5)=1, and ab=144*5=720 indeed. But a=144>5, so this would not satisfy a < b, therefore this assignment would not be included in the count. So how do we reconcile the count. If the number of coprime assignments (with order) is 2^3=8, but how many of those assignments result in a < b? So for 720, with prime factors 2,3,5:The 8 assignments:1. Assign none to a => a=1, b=720 (1 <720: yes)2. Assign 2 to a =>a=16, b=45 (16 <45: yes)3. Assign 3to a =>a=9,b=80 (9 <80: yes)4. Assign5 to a =>a=5, b=144 (5 <144: yes)5. Assign 2 and3to a => a=144, b=5 (a >b: no)6. Assign 2 and5 to a => a=16*5=80, b=9 (80 >9: no)7. Assign3and5 to a => a=9*5=45, b=16 (45>16: no)8. Assign all three primes to a =>a=720, b=1 (a> b: no)Therefore, out of 8 assignments, 4 result in a< b and 4 in a>b. So when 720 is not a square, which it is not (since exponent of 5 is 1, odd), thus the case a=b is not possible, hence the number of ordered coprime pairs with a < b is half the total, which is 4. So in general, the answer seems to be half of the number of coprime ordered pairs.Therefore, again, number for 20! would be 2^8/2 =128. So128 coprime pairs(a,b) with a <b. However, perhaps there's a problem here. Wait, in the case when assignments are made for each prime, but, in the original problem, the numerator and denominator are reduced to lowest terms. Which in original fraction it's not required that the numerator and denominator multiply to 20!; rather, the reduced fraction \(\frac{a}{b}\) has the product a*b which is 20 factorial. Wait, that needs to be resolved. Wait, perhaps that's the confusion. The original number is some fraction as part of between 0 and1. When written in lowest terms, numerator a and denominator b must satisfy that. However, a different way to phrase this is: a is numerator and b is denominator in reduced form (so gcd(a, b)=1), and when you compute the product a*b, this product is equal to 20!. Hence, the question is equivalent to: how many pairs (a, b) are there with gcd(a,b)=1, 0 <a < b, and a *b = 20!. Which translates as, (since 20! is fixed and a*b=20!), how many ways can 20! be split into two coprime factors a and b, both positive integers, a <b.And this number is equal to 2^{k -1},where k is the number of distinct prime factors of 20!.But since the number of distinct prime factors of 20! is 8 (since primes 2, 3, 5,7, 11,13,17,19).Hence, 2^{8 -1}= 128. Hence, result. Wait, this would be 2^{k-1}=128. Alternatively, total number of subsets is 2^8=256, each ordered pairs. Taking half because a< b hence 256/2=128. So confirms. But let me check with the 6! example. For 6!, which had instead three distinct prime factors (2,3,5), number of coprime pairs where a < b is 4 (by explicit count above). But 2^{3-1}= 4 correct. Hence, Therefore, the formula is 2^{k -1}.Hence, for 20! with8 prime factors, number of coprime pairs with a <b and a*b=20! is 2^{8-1}=128, which seems correct.But Wait, what about multiplicity. For example, 1and 720 were co-prime (a=1, b=720). Then you had16 and 45, and 9 and 80, and 5 and 144. How is it multiplicative, related to the number of prime factors?Yes, but 6! is 720, and had three prime factors. The number of coprime pairs a < b where a*b=720 is 4, which is 2^{3-1}=4.Similarly, if n is a square-free number, which 720 is not square -free (as 720=16*45=24^2*5).But in general, n! will have prime factors multiplicities.But our problem states n=20. So 20!.Regardless of the multiplicities (since in order for a and b to be coprime, the entire allocation of each prime has to go to one factor). Therefore, how primes are assigned has a per-prime dependence.Wait, so since for each prime, regardless of exponent (even or odd), we must give it entirely to a or b,which leads to these coprime pairs. So seems that the number is, in general, 2^{k} ordered coprime pairs (a,b), and because n! is not a square (unless for example primes with even exponents), it depends.

...(Omit)... but depending on which primes.But here is the crux:instead of trying to group, this is again difficult to calculate by separate method; but via generating the coprimality by prime division, and noting in N! is fixed, so that the only way to a*b=N!:Since a and b are coprime.Hence equivalence between choices over subsets of primes.Hence, since ordered coprimes, 2^pairs.Each with a Unique Subsets of primes assigned to a.Hence, since ordering for each subset, half will be such that the a is less than b.Hence if done for k primes, because N! is not a square, then exactly half of those subsets a*a less than N!, that is, a < sqrt(N!).But for all possible assignments, the total number of cases where a < b should be equal to 2^{k-1}as reasoned before.Therefore, the previous answer 128must be correct, hence The answer would be 128.But before closing, verifying with our 6! example was key.In the 6! example (k=3 primes), number of coprime pairs with a < b=4=2^{3-1}.And with actual calculation matching.Therefore, Conclusion:The number of rational numbers between0 and 1 such that when written in lowest terms as a/b, the product of numerator and denominator is 20!=1·2·3...·20 is given by 2^{number of prime factors of 20! -1} =2^{8-1}=128.**Final Answer**\boxed{128}<lim_startl>answerAnswer: Given a rational number between 0 and 1, we need to write it as a fraction in lowest terms \(\frac{a}{b}\) and calculate the product of the resulting numerator and denominator. We need to determine how many such rational numbers exist where the product \(a \times b = 20!\).1. **Prime Factorization of 20!**: - The prime factors of 20! are 2, 3, 5, 7, 11, 13, 17, and 19. These are 8 distinct primes.2. **Coprime Pairs**: - For \(a \times b = 20!\) and \(\gcd(a, b) = 1\), each prime factor of20! must be assigned entirely to either \(a\) or \(b\). This results in \(2^8= 256\) coprime pairs \((a, b)\).3. **Counting Pairs \(a < b\)**: - Since \(a\) and \(b\) are coprime and \(a \times = 20!\), each pair \((a,b)\) and \((b, a)\) are distinct. To find pairs where\(a < b\), we divide the total number of pairs by 2, excluding the case \(a = b\) (which is impossible since 20! is not a square).4. **Result**: - The number of coprime pairs \((a, b)\) with \(a < b\) is \(\frac{256}{2} = 128\).Thus, the number of rational numbers between 0 and 1for which the product of the numerator and denominator in lowest terms is \(20!\)is \(\boxed{128}\).<lim_endl>

