# OpenReview forum: "AIR: Post-training Data Selection for Reasoning via Attention Head Influence"
_ICML.cc/2026/Conference — ICML 2026 regular_

### Official Review · Reviewer_m6wN · 2026-02-27

**Soundness:** 3
**Presentation:** 3
**Significance:** 3
**Originality:** 3
**Overall Recommendation:** 5
**Confidence:** 4

**Summary:**

This paper proposed AIR score, which can be used for selecting reasoning steps/samples for post-training distillation. If the token-level losses change significantly once the copy heads in the LLM are masked, then the corresponding tokens will be flagged as a reasoning-relevant token. The AIR framework is carefully verified to be effective in multiple benchmarks.

**Compliance With Llm Reviewing Policy:**

Affirmed.

**Final Justification:**

In the rebuttal phase, the authors address my concerns about motivation, other baselines, and the significance of the results through extensive experiments. I'm willing to see the paper being accepted.

**Key Questions For Authors:**

1. In Line 149, $w_{1:t-1}$ once confused me, i think $w_{n+1, t}$ may be better.
2. What is your opinion on mechanistic tools other than copy head identification, like Sparse Auto-encoder[1, 2]?
3. Other problems are listed in Weaknesses already.

[1] Controllable LLM Reasoning via Sparse Autoencoder‑Based Steering.
[2] Resa: Efficient Reasoning Models via SAEs.

**Limitations:**

It would be beneficial for the authors to add a section discussing the limitations of this paper, such as those related to LLM and dataset selection.

**Strengths And Weaknesses:**

**Strengths**
1.	Good Representation about the details of the proposed AIR & AIR scores.
2.	Detailed evaluation on the several reasoning benchmarks and case study for AIR framework.
3.	Important research problem. Considering that data synthesis makes progress quickly, I’d love to see investigations on data selection.
**Weaknesses**
1.	The motivation of AIR framework should be clarified more precisely. Specifically, what is the relation between ``copy-paste’’ operation and reasoning? Copying can be a small part of the basic reasoning capabilities in my opinion, such as retrieving previously used variable names during code generation. In Math/Knowledge Reasoning, however, it seems difficult to obtain correct answers only with copying operation, where relying on AIR scores may hinders the creativity or other reasoning skills. I notice the author conduct analysis on the selected samples, what I want is a intuitive explanation. In this sense, this point hinders the AIR framework distinguish with other heuristic approaches, I would raise my score if this problem is addressed :)
2.	I’m curious about the performance of (process) reward models in data selection. The AIR scores, in my opinion, can somehow be viewed as a heuristic reward score also.
3.	In s1K Setting(Distilling from Gemini), the proposed method does not outperform baseline s1K, which may limits the overall significance.  Also, in Figure 2, the performance seems sensitive to hyper-parameters, the accuracy drops significantly (Reporting Pass@5/10 may be effective to resolve my concerns).

---

> ### Author Rebuttal · Authors · 2026-03-31
>
> We sincerely thank the reviewer for the insightful and constructive comments.
>
> **W1:** We thank the reviewer for raising this question! Intuitively, the 'copy-paste' mechanism in LLMs transcends mere copying; it represents the model's fundamental ability to maintain factual integrity. Just as a human solving a complex math problem refers back to the problem statement and prior equations, an LLM during reasoning often refers back to the prompt and its previously generated context. Therefore, rather than hindering creativity, the copying operation anchors exploratory steps and preventing logical fallacies. Our perturbation experiments (see response to Reviewer Gfsq Q1) also support this perspective. Furthermore, research [1] confirms that masking retrieval heads in reasoning-intensive tasks drastically degrades performance. This occurs because, without effective retrieval, the model tends to 'ignore' crucial conditions, leading to erroneous reasoning, whereas tasks where the model directly generates answers using its intrinsic knowledge are less affected by such masking. This mechanistic foundation distinguishes AIR from simple heuristic methods, confirming that our method can effectively prioritize high-quality reasoning data that genuinely trains the model to ingest context and execute rigorous reasoning.
>
> [1] Retrieval head mechanistically explains long context factuality
>
> **W2:** We appreciate you highlighting this relevant research. Conceptually, step-level AIR aligns with process reward models (PRMs) like ReasonFlux-PRM [1], offering fine-grained supervision for long reasoning. Unlike PRMs relying on costly annotations, AIR is entirely unsupervised and training-free. Instead of training PRM or using superficial heuristics (e.g., reflection token frequency), AIR exploits foundation models' intrinsic mechanisms via interventions on critical retrieval heads. We also conduct performance tests on the step-weighted training of ReasonFlux-PRM-7b and AIR-step. As shown below, despite being unsupervised and training-free, AIR-Step outperforms ReasonFlux-PRM, further demonstrating the effectiveness of our method.
>
> | Method | AIME24 | MATH | GPQA |
> | :--- | :--- | :--- | :--- |
> | ReasonFlux | 60.00 | 94.60 | 61.11 |
> | AIR-Step | 66.70 | 95.60 | 65.66 |
>
> [1] ReasonFlux-PRM: Trajectory-Aware PRMs for Long Chain-of-Thought Reasoning in LLMs
>
>
> **W3:** We appreciate your constructive feedback! While AIR-Sample does not outperform the s1K baseline under the Gemini-distilled setting, it is worth noting the differences in their methodologies: s1K relies on human-assisted selection and complex filtering, whereas AIR is training-free and fully automated. Achieving competitive performance against human-curated datasets using a single automated metric highlights our method's efficacy. Furthermore, on higher-quality trajectories (DeepSeek-R1 setting), AIR comprehensively surpasses s1K-1.1. Regarding Figure 2, your observation is astute: as discussed in the manuscript, an excessive selection ratio introduces redundant steps, while an overly high weight multiplier ($\alpha > 5$) distorts the local loss landscape.
>
> Finally, we value your suggestion to report Pass@K, measuring intrinsic reasoning stability. Evaluations of the AIR-step model on AIME24/25 (temperature $T=1$) are detailed below and will be included in the revised appendix to address robustness.
>
> | Metric | AIME 2024 (s1k-1.1) | AIME 2024 (AIR-step) | AIME 2025 (s1k-1.1) | AIME 2025 (AIR-step) |
> | :--- | :--- | :--- | :--- | :--- |
> | Pass@4 | 74.19 | 82.00 | 61.10 | 66.33 |
> | Pass@8 | 76.67 | 90.00 | 66.67 | 73.33 |
> | AVG@8 | 56.25 | 60.42 | 46.25 | 47.50 |
>
> **Q1:** Thank you for the advice. To ensure strict mathematical consistency with the current decoding step $t$, we will revise $w_{n+1:n+t-1}$ in the revision. Equation (1) and surrounding text will also be updated to make token positioning crystal clear.
>
>
> **Q2:** We thank the reviewer for pointing out the relevant literature [1,2]. We fully agree that Sparse Autoencoders (SAEs) [1, 2] are also powerful tools for mechanistic interpretability. However, AIR and SAEs operate on different architectural structures and serve distinct downstream objectives. Mechanistically, SAEs target the hidden states of LLMs, regulating model behavior by manipulating these representations. In contrast, AIR focuses on exploiting retrieval heads which are specific structures that exerts a strong influence on a model's reasoning capabilities,and is designed explicitly for data selection. By leveraging the unsupervised AIR scoring mechanism, we can evaluate reasoning trajectories without requiring any external training or structural modifications to the model. Given these distinct focal points, SAEs and AIR are highly complementary and hold great potential for mutual integration in future research. Thanks for your great comments again.

---

> > ### Author Rebuttal · Reviewer_m6wN · 2026-04-01
> >
> > I have received point to point response from the author. Most of my concerns, including W2&W3, have been largely addressed by the new experimental results. Also, my questions obtained point to point answers. I'm going to raise my score to weak accept for the valuable effort.
> >
> > The reason why I don't raise my score higher is that, the response for W1 is not sufficient to me. Specifically, the author's clarification is limited to the necessity of the 'copy-paste' mechanism, rather than its relationship with and impact on other forms of reasoning, such as creativity.
> >
> > ---
> > The following rebuttal reply further address my concern about W1. I will raise my score to 5 to support its acceptance.

---

> > > ### Author Response · Authors · 2026-04-04
> > >
> > > We sincerely thank the reviewer for recognizing our work and raising the score. Here, we try to clarify the relationship and impact of the 'copy-paste' mechanism on other forms of reasoning, especially creativity.
> > >
> > > To better address this question, we conducted an additional analysis of reasoning characteristics by examining the frequency of representative connective words in the curated training dataset (AIR-Sample) compared to the baseline dataset (s1K-1.1). These connective words correspond to different reasoning behaviors, including exploratory, planning, verification, and causal reasoning. For example, exploratory connectives such as “alternatively,” “another way,” “maybe,” and “perhaps” are commonly associated with divergent thinking and creative exploration. Detailed definitions and computation methods for all connective types are provided in Appendix B.4. The results are summarized below:
> > >
> > > | Metric (All in %) | s1K-1.1 (Baseline) | AIR-Sample (Ours) |
> > > | :---------------- | :----------------- | :---------------- |
> > > | Exploratory       | 1.58               | 1.75              |
> > > | Planning          | 0.75               | 0.88              |
> > > | Verification      | 0.28               | 0.33              |
> > > | Causal            | 2.70               | 2.80              |
> > > | Contrast          | 1.31               | 1.29              |
> > > | Correction        | 1.04               | 0.97              |
> > >
> > >
> > > These metrics demonstrate that the AIR dataset preserves and even amplifies the key reasoning traits. In particular, the increase in Exploratory and Planning markers indicates that the selected trajectories retain the ability to explore alternative solutions and organize intermediate steps when reasoning becomes non-trivial. As shown in Table 4 of our original manuscript, the output analysis of the AIR-step model exhibits a similar trend.
> > >
> > > In addition, we evaluate text diversity using Distinct-3, which measures the proportion of unique 3-grams in the generated text. A higher Distinct-3 score indicates greater lexical diversity and variation in reasoning expressions. As shown below, the significant increase in Distinct-3 (+2.83%) confirms that the curated trajectories maintain a diverse set of reasoning patterns rather than collapsing to repetitive formulations. This dual validation at both the dataset and model levels demonstrates that our method naturally curates and transfers a diverse range of complex reasoning capabilities.
> > >
> > >
> > > | Metric (All in %) | s1K-1.1 (Baseline) | AIR-Sample (Ours) |
> > > | :---------------- | :----------------- | :---------------- |
> > > | Distinct-3        | 72.95              | 75.78             |
> > >
> > > From a mechanistic perspective, AIR does not reward superficial copying. Instead, it captures a more fundamental capability: grounding reasoning in previously established context. This grounding process naturally involves skills such as verification and reflection, while also providing a stable basis for more reliable higher-level reasoning, including creative exploration and compositional problem solving, particularly in domains such as mathematical and knowledge-intensive reasoning.
> > >
> > > This intuition is also reflected in the step-level visualizations provided in the paper (e.g., the cases in Figure 5 and Appendix C). In Figure 5, the reasoning trajectory exhibits advanced creative exploration and self-correction behavior: the model actively pauses routine deduction to ingeniously construct a smaller sub-problem ($n=6$) to verify its hypothesis. To execute such delicate, higher-order reasoning steps, the model needs to precisely retrieve and adhere to the constraints already established in the context, such as the $a < b$ limitation. If the support of the 'copy-paste' capability is lost at this moment, the model will forget its premises. Even if the model attempts to make exploratory leaps, transitions, or new hypotheses, these ungrounded higher-order actions are highly likely to be incorrect and will degenerate into logical fractures and hallucinations, resulting in a notable loss discrepancy during our masked evaluation. It is precisely because these advanced reasoning actions are sensitive to the 'copy-paste' mechanism that AIR, by capturing this loss discrepancy, naturally filters out the high-quality data containing creativity and other forms of reasoning.
> > >
> > > Taken together, these findings suggest that AIR does not suppress creativity. Rather, it supports more reliable and structured forms of creative and compositional reasoning. We will incorporate this clarification into the main paper to further strengthen the conceptual presentation of AIR. We hope the above clarification helps address your concerns. We thank you again for your valuable comments.

---

### Official Review · Reviewer_kvgG · 2026-03-12

**Soundness:** 2
**Presentation:** 3
**Significance:** 3
**Originality:** 3
**Overall Recommendation:** 4
**Confidence:** 4

**Summary:**

This paper proposes AIR (Attention Influence for Reasoning), a post-training data selection and reweighting method. The core idea is based on a mechanistic intervention: identifying "retrieval heads" (heads that perform token-level copying) and measuring the model's reliance on them. By masking these heads (setting attention to uniform) to create a weakened reference, the authors use the resulting loss gap as a proxy for data quality. This signal is applied to both sample-level filtering (AIR-Samp) and step-level weighting (AIR-Step). The method is tested on reasoning benchmarks like AIME and MATH, showing improved efficiency over random-masking baselines.

**Compliance With Llm Reviewing Policy:**

Affirmed.

**Final Justification:**

My concerns have been adequately addressed.

**Key Questions For Authors:**

1. Is AIR selecting "logic" or "copies"? I suggest you provide n-gram overlap stats for Top vs. Bottom AIR steps. If you could human-label 200 steps to see if AIR scores "logical derivations" higher than "problem restatements," it would significantly strengthen the paper.

2. Does the intervention operator matter? If you use zeroing-out or dropout instead of uniform-attention masking, do the AIR rankings and downstream gains remain consistent?

3. Better matched controls: Instead of just random masking (RAIR), can you mask non-retrieval heads that have similar average attention magnitude? This would prove the effect is specific to the retrieval function, not just any "important" head.

4. Functional Alignment: When using a small model to score for a large one, do the identified heads actually perform the same "retrieval" task in both models?

5. Experiments: There are experiments on AIR samples and AIR-steps+s1k-1.1; why not carry out experiments on AIR samples + AIR steps to see the performance of the combination?

**Limitations:**

See weaknesses part.

**Strengths And Weaknesses:**

**Strengths**

Practically efficient: The scoring is training-free and only requires forward passes. Avoiding external LLM graders or training separate reward models makes it very appealing for large-scale data pipelines.

Versatile utility: Using the same signal for both dataset pruning and in-training step weighting is a pragmatic design that fits well into standard post-training recipes.

Principled motivation: Targeting a specific functional sub-system (retrieval heads) is a more grounded approach than using generic surface-level metrics like perplexity or length.

**Weaknesses**

Risk of "Lexical Copying" bias: Because retrieval heads are defined by token-level copying, there is a serious risk that AIR simply favors steps that heavily restate the prompt or repeat context. In complex reasoning, copying is not the same as thinking. Without evidence that AIR selects "logical pivots" rather than "lexical echoes," the performance gains might just come from reducing transcription errors rather than enhancing reasoning.

Confounding from structural perturbation: Forcing attention to a uniform distribution is a violent structural change. The resulting loss increase might just be the model reacting to an "unnatural" internal state (distribution shift) rather than a genuine causal reliance on that specific mechanism.

Unverified cross-scale transfer: The paper claims that identifying heads in a small model (e.g., 1.5B) works for a larger target (32B). While plausible, the paper lacks a rigorous check of whether these heads actually align in function or location across different model sizes.

---

> ### Author Rebuttal · Authors · 2026-03-31
>
> Thank you for your insightful comments and suggestions.
>
> **Q1 \& W1**: We appreciate the reviewer's suggestion and are pleased to provide the requested experimental evidence below. The 3-gram overlap statistics demonstrate that while the top-scoring steps exhibit a slightly higher overlap rate than the bottom-scoring steps, the absolute values remain exceedingly low (<3%). This confirms that these steps are not mere verbatim copies; rather, they ensure logical coherence by maintaining the consistency of key information. We also provide empirical evidence at the sample level in our response to Reviewer 55fH (W2).
>
> | Group         | 3-Gram Overlap (%) |
> | :--- | :--- |
> | Bottom AIR Step   | 1.12               |
> | Top AIR Step      | 2.14               |
>
> Furthermore, following your great suggestions, we employ Gemini Pro as an automatic evaluator to classify 200 sampled steps and compute the average AIR-Step scores for each group, as manual annotation is both time-consuming and inherently subjective. The results below show that logical derivation steps achieve significantly higher mean scores than uninformative steps. Additionally, the case studies provided in our appendix (B.3 & C) further corroborate this quantitative finding. For a more detailed conceptual discussion, please refer to our response to Reviewer m6wN (W1).
>
> | Group  | Mean AIR Score     |
> | :--- | :--- |
> | Uninformative step | 0.1575             |
> | Logical step       | 0.6981             |
>
>
> **Q2 \& W2**: We appreciate the reviewer’s constructive question. To evaluate the robustness, we conducted additional experiments by implementing zeroing-out as an alternative to our default setting. The step-level results below demonstrate that the performance improvements remain consistent, confirming that the causal importance of reasoning-critical steps identified by AIR is not sensitive to the specific mathematical protocol used to disable head influence. This stability underscores that AIR's primary value lies in its principled identification of reasoning-intensive data through mechanistic interpretability, rather than the specific technicalities of the intervention.
>
> |Method| AIME24 | MATH500 | GPQA Diam. |
> |:---| :--- | :--- | :--- |
> | AIR (Zero)| 60.00 | 95.2 | 64.64 |
> | AIR (Uniform) | 66.70 | 95.60 | 65.66 |
>
> **Q3**: We appreciate the reviewer’s suggestion to incorporate a more rigorous control by masking non-retrieval heads with similar average attention magnitudes (MAIR). Our updated experimental results across three benchmarks demonstrate the specificity of retrieval heads: while masking high-magnitude non-retrieval heads (MAIR) yields a marginal improvement over random masking (RAIR), it is consistently outperformed by AIR-Step. This performance gap confirms that the reasoning-critical steps identified by AIR uniquely depend on the specialized mechanism provided by retrieval heads rather than simple attention prominence.
>
> |Method| AIME24 | MATH500 | GPQA Diam. |
> | :--- | :--- | :--- | :--- |
> | AIR | 66.70 | 95.60 | 65.66 |
> | MAIR | 60.00 | 94.40 | 64.14 |
> | RAIR | 53.33 | 94.60 | 60.10 |
>
> **Q4 \& W3**: We greatly appreciate your valuable insights! We agree that the functional alignment between the scoring model and the target model is crucial for the cross-scale effectiveness of AIR. Recent theoretical research on retrieval heads [1] demonstrates that "retrieval heads" exhibit significant intrinsic consistency across different scales within the same model family. This study finds that retrieval heads within the same family are strongly correlated, sharing the same set of retrieval heads and performing identical tasks. This cross-scale intrinsic consistency enables smaller models to serve as reliable "guides" for larger ones. Empirically, we validated this alignment in Table 2, where reference models of varying scales (1.5B and 7B) both significantly outperformed the random baselines when scoring data for the 32B model. This proves that the fundamental identification of reasoning-critical steps remains robust across different scales.
>
> [1] Retrieval Head Mechanistically Explains Long Context Factuality
>
> **Q5**: Thank you for the valuable suggestion. We agree that combining AIR-Sample and AIR-Step could be an interesting experiment to evaluate potential synergistic effects. In our current study, we focused on evaluating each method individually to clearly isolate its respective contributions. While this combination was part of our original plan, we could not complete the full evaluation within the available timeframe. We are conducting these experiments and commit to including the results in the final version.

---

> > ### Author Rebuttal · Reviewer_kvgG · 2026-04-03
> >
> > Thank you for your detailed response. The response has solved my concern and it is better to include them in the revised version. I will maintain my score.

---

> > > ### Author Response · Authors · 2026-04-04
> > >
> > > Thank you sincerely for your time, constructive feedback, and positive recognition of our rebuttal. We will ensure that these new results and analyses are included in the revised version. We would greatly appreciate any further consideration in your final assessment, and we remain truly thankful for your positive evaluation and valuable insights.

---

### Official Review · Reviewer_Gfsq · 2026-03-13

**Soundness:** 2
**Presentation:** 3
**Significance:** 3
**Originality:** 3
**Overall Recommendation:** 3
**Confidence:** 3

**Summary:**

The paper focuses on data selection in post-training. The authors propose the Attention Influence for Reasoning (AIR) framework, which quantifies the importance of data based on the mechanistic behavior of retrieval attention heads. The AIR score effectively evaluates both sample-level quality and step-level quality. More importantly, the authors show strong cross-scale transferability: by efficiently computing the AIR scores using a small model (e.g., 7B), high-quality data can be selected to successfully enhance the reasoning capabilities and training efficiency of a much larger model (e.g., 32B).

**Compliance With Llm Reviewing Policy:**

Affirmed.

**Final Justification:**

My concerns have been addressed.

**Key Questions For Authors:**

1.  The AIR score measures the importance of the data. However, high influence doese not necessarily equate to high quality. If a data point is incorrect or harmful, it may still exert a huge impact on the gap $delta$ between $l_(\theta_{\text{base}})$ and $l_(\theta_{\text{ref}})$. How do we avoid this? Or do you have some experiments on the attack of this method? Whether it is robust or not?

2. Should data selection criteria vary across different phases of LLM training? Equivalently, is the definition of 'high-quality data' inherently stage-dependent?

3. Tiny question: Have you tried AIME25 instead of AIME24 for table 2?

**Limitations:**

yes

**Strengths And Weaknesses:**

Strengths:
1. The paper addresses a high-impact topic with clear relevance to current industry needs. It's very important!
2. The proposed method is intuitive, consistent, and effectively addresses the complexities of the problem.
3. The simulation results are well-documented, demonstrating a thorough exploration

Weakness:
1. See the questions.

---

> ### Author Rebuttal · Authors · 2026-03-31
>
> We sincerely thank the reviewer for your valuable comments and suggestions.
>
> > **Q1：High influence doese not necessarily equate to high quality.**
>
> We sincerely thank the reviewer for their profound insights regarding the distinction between data influence and data quality. To address concerns regarding potential "bad data attacks," we conducted supplementary robustness experiments using 200 samples: 100 original valid reasoning trajectories and 100 perturbed samples where numbers were randomly removed or altered. As shown below, AIR effectively distinguishes between these groups: the average AIR score for the valid group ($0.310$) is more than double that of the perturbed group ($0.144$). Notably, when filtering the Top-20% of samples by AIR, 97.5% were original valid samples, strongly indicating that AIR prioritizes data exhibiting both causal dependencies and coherent reasoning. This behavior is consistent with the underlying mechanism of AIR: while erroneous data may still influence the raw loss, their broken logical structure fails to activate the specialized retrieval heads that are critical for maintaining logical coherence. As a result, such samples receive substantially lower AIR scores.
>
> | Group | #Samples | Avg. AIR Score |
> |-----------------|----------|----------------|
> | Perturbed   | 100      | 0.144   |
> | Valid       | 100      | 0.310   |
>
> Furthermore, prior work on reasoning distillation suggests that as long as reasoning paths remain coherent, the negative impact of a small proportion of incorrect data is limited. For example, models fine-tuned on datasets with only 85.4% accuracy still significantly outperform Zero-RL baselines, indicating that models learn higher-level cognitive behaviors (e.g., multi-perspective reasoning) rather than merely memorizing correct answers. By prioritizing data with proper logical structure, AIR provides an additional layer of robustness against harmful or illogical data.
>
> [1] Why Distillation can Outperform Zero-RL: The Role of Flexible Reasoning
>
> > **Q2：Should data selection criteria vary across different phases of LLM training? Equivalently, is the definition of 'high-quality data' inherently stage-dependent?**
>
> Thank you for raising the theoretical question regarding the stage-dependency of data quality. Although the definition of quality may emphasize different aspects, we believe that the fundamental mechanism of effective reasoning remains consistent across training stages. This is reflected in RHO-1, which transitions from a pretraining strategy to a highly effective baseline in our post-training experiments. Accordingly, we believe that the AIR score is essentially also applicable to pretraining data selection, as it targets the retrieval heads present in models. Although we prioritized post-training applications due to the extremely high computational demands of large-scale pretraining, the mechanistic signals captured by AIR represent a stage-independent criterion for identifying data that can drive core reasoning capabilities.
>
> > **Q3: Have you tried AIME25 instead of AIME24 for table 2?**
>
> Thank you for the comments. Following the $s1$ experimental protocol, our initial evaluation in Table 2 focused on AIME 2024, MATH500 and GPQA Diam. To facilitate a clearer comparison, we provide the supplementary results for AIME 2025. As shown below, the results on AIME 2025 consistently align with our primary findings. First,models with different size can all achieve performance gains. Second, performance degradation observed with random masking (RAIR) in both sample- and step-level settings,these directly confirms that AIR's effectiveness stems from targeting reasoning-critical retrieval heads, rather than merely introducing random noise.
>
> | Method | AIME 2025 |
> | :----- | :----- |
> | s1K | 26.70 |
> | RAIR-Sample (1.5B)| 20.00 |
> | AIR-Sample (1.5B) | 24.70 |
> | AIR-Sample (7B) | 23.33 |
> | s1K-1.1 | 50.00 |
> | RAIR-Step (7B) | 36.70 |
> | AIR-Step (1.5B) | 50.00 |
> | AIR-Step (7B) | 53.33 |

---

> > ### Author Rebuttal · Reviewer_Gfsq · 2026-04-03
> >
> > Thank you to the author for the detailed response.

---

> > > ### Author Response · Authors · 2026-04-04
> > >
> > > Thank you sincerely for your time, constructive feedback, and positive recognition of our rebuttal. We are delighted to know that your concerns have been fully resolved. We would greatly appreciate any further consideration in your final assessment, and we remain truly thankful for your evaluation and valuable insights.

---

### Official Review · Reviewer_55fH · 2026-03-13

**Soundness:** 2
**Presentation:** 3
**Significance:** 2
**Originality:** 3
**Overall Recommendation:** 4
**Confidence:** 3

**Summary:**

This paper proposes AIR, a framework for post-training data selection aimed at efficiently distilling multi-step reasoning abilities. AIR is motivated by mechanistic interpretability findings that certain Transformer retrieval attention heads are crucial for refer-back behavior in chain-of-thought reasoning. The method first identifies these reasoning-critical heads via a retrieval score, then constructs a weakened reference model by masking the identified heads, and finally defines an attention influence score as the loss divergence between the base model and the weakened model. This score is aggregated to support both sample-level selection and step-level weighting.

**Compliance With Llm Reviewing Policy:**

Affirmed.

**Final Justification:**

My concerns have been adequately addressed.

**Key Questions For Authors:**

See Weaknesses

**Limitations:**

See Weaknesses

**Strengths And Weaknesses:**

Strengths：
1. The paper is clearly written, with a clear problem statement and a logically organized presentation.
2. The experimental evaluations show consistent and meaningful performance gains across multiple reasoning benchmarks.

Weaknesses：
1. AIR scores may systematically favor longer solutions with multiple rounds of self-reflection/verification/rollback, since such trajectories frequently re-reference earlier conditions and intermediate conclusions, which are exactly the tokens likely to rely on retrieval heads. As a result, the reported gains might be partially explained by selecting longer reflective CoTs rather than AIR capturing a uniquely mechanistic signal of reasoning criticality.
2. To validate that AIR provides signal beyond surface redundancy, it would be informative to include a baseline that selects samples with the highest token/phrase repetition rate, under the same category quotas and budget (e.g., top-1K). If such a redundancy-based selection yields similar improvements, it would suggest AIR’s gains might largely stem from verbosity/reflection patterns. If not, it would strengthen the claim that AIR captures non-trivial mechanistic dependence rather than superficial properties.
3. The post-training is demonstrated mainly for Qwen2.5-32B following the s1 protocol. It remains unclear how robust AIR is on more recent or different reasoning-oriented model families (e.g., Qwen3).

---

> ### Author Rebuttal · Authors · 2026-03-31
>
> We sincerely thank the reviewer for your valuable comments and suggestions.
>
> > **W1：AIR scores gains might be partially explained by selecting longer reflective CoTs**
>
> Thank you for this insightful observation. We address this concern from three complementary perspectives:
>
> (1) *AIR does not favor longer training samples.* To directly examine whether AIR systematically prefers longer reflective CoTs, we conducted a supplementary analysis of the average lengths across different constructed datasets in the s1K-1.1 setting. As shown below, AIR-Sample dataset has a significantly shorter average length (23,809) compared to the human-curated s1K-1.1 dataset (29,220). This provides clear evidence that AIR does not bias toward longer reasoning trajectories.
>
> | Method | Average Length |
> |-----------------|------------------------|
> | s1K-1.1         | 29,220                 |
> | AIR-Sample      | 23,809                 |
>
>
> (2) *AIR leads to more concise yet more accurate reasoning at inference time.* As shown in Table 4 (main paper), models trained on AIR-step data achieve higher accuracy and exhibit richer reasoning connectives while generating more concise responses (10,537 vs. 10,652 tokens) compared to those trained on s1K-1.1. This further rules out the possibility that performance gains arise from longer or more verbose reasoning chains.
>
> (3) *AIR outperforms explicit length-based selection.* If length were the key factor, a simple “Length-1K” baseline should perform competitively. However, as reported in Table 1 (main paper), the Length-1K baseline consistently underperforms AIR across most benchmarks. This demonstrates that AIR captures a different signal beyond superficial length.
>
> Overall, these results collectively indicate that the performance gains from AIR are unlikely to be solely due to selecting longer reflective CoTs and are instead associated with its ability to identify more informative reasoning steps.
>
>
> > **W2: baseline that selects samples with the highest token/phrase repetition rate.**
>
> We sincerely thank the reviewer for this highly inspiring suggestion. To verify whether AIR captures deep mechanistic dependencies rather than surface-level lexical redundancy, we supplemented a baseline experiment as per your advice. Specifically, we utilized a 3-gram metric to calculate the repetition rate for each sample in the data pool and selected those with the highest redundancy for fine-tuning, while under the same category quotas and budget as AIR-Sample.
>
> As shown below, the performance of this redundancy baseline is lower than that of AIR, especially in high-difficulty reasoning tasks (such as AIME 2024). This result demonstrates that AIR’s performance gains do not stem from surface redundancy, but rather from its ability to select reasoning data through the effective utilization of retrieval mechanisms, which are essential for maintaining logical rigor. We will include this experiment and the corresponding discussion in the revision.
>
> | Method | AIME 2024  | MATH500 | GPQA Diamond | Average |
> | :--- | :--- | :--- | :--- | :--- |
> | 3-Gram | 30.00 |  89.40 | 43.43 | 47.38 |
> | AIR-Sample  | 50.00 | 90.80 | 55.00 | 54.78 |
>
> > **W3: It remains unclear how robust AIR is on more recent or different reasoning-oriented model families (e.g., Qwen3)**.
>
> Thank you for the forward-looking suggestion regarding newer model families like Qwen3. Although our current evaluation strictly follows the $s1$ protocol using Qwen2.5-32B, the robustness of AIR is grounded in the "retrieval head" mechanism inherent to the Transformer architecture. Empirical evidence in [1] also suggests that these heads consistently constitute approximately 5% of the attention heads across diverse models and scales.
>
> Furthermore, we also test the effectiveness of our AIR for Qwen3-8B-Think model. As shown below, AIR-Sample continues to match or outperform the human-curated s1K-1.1 baseline, further validating that AIR is a generalizable mechanism rather than overfitting to a specific model family. We will include these supplementary results in the revison to further demonstrate AIR's generalizability.
>
> | Method | AIME 2024 | GPQA Diamond | MATH500 | Average |
> | :--- | :--- | :--- | :--- |:---|
> | s1k-1.1 | 66.67 | 57.07 | 93.8 |72.51|
> | AIR-Sample | 66.67 | 60.60 | 94.4 |73.89|
>
>
>
> [1] Retrieval head mechanistically explainslong context factuality.

---

> > ### Author Rebuttal · Reviewer_55fH · 2026-04-03
> >
> > Thank you for your detailed response to my comments. I appreciate the time and effort you have put into addressing the raised concerns. The clarifications and revisions are satisfactory, and I am happy to raise my score.

---

> > > ### Author Response · Authors · 2026-04-04
> > >
> > > Thank you sincerely for your time, constructive feedback, and positive recognition of our rebuttal. We will ensure that these new results and analyses are included in the revised version. We are again truly grateful for your positive evaluation and valuable comments.

---

### Decision · Program_Chairs · 2026-04-30

**Decision:**

Accept (regular)

**Comment:**

This paper proposes AIR, a post-training data selection framework for efficiently distilling multi-step reasoning abilities by leveraging attention influence scores derived from reasoning-related retrieval heads. The reviewers found the paper clearly written and well organized, and viewed the problem as timely and important. They also noted that the method is practically appealing, especially because it is training-free, computationally efficient, and can support both sample-level selection and step-level weighting.

During the rebuttal and discussion period, the reviewers generally responded positively to the authors’ additional analyses and clarifications. Reviewers found that the new ablation studies, analytical experiments, and additional baseline comparisons substantially improved the empirical support for the paper, and helped strengthen the case that the gains are not simply due to superficial heuristics. The paper’s Presentation and Originality were consistently assessed positively, and several reviewers considered the mechanistically motivated data selection perspective to be a meaningful contribution that others may build on. At the same time, some concerns remained regarding the exact connection between retrieval behavior and reasoning quality, possible bias toward lexical copying or longer reflective chains, and the extent to which the cross-scale transfer claims are fully validated. Still, the overall discussion suggests that these concerns do not outweigh the paper’s practical contribution, solid empirical results, and clear relevance to reasoning data selection.

In light of these feedback, we recommend acceptance. While the paper has some limitations and would benefit from further validation in future work, the reviewers found that it offers a useful and timely contribution to an active area, with sufficient technical quality and empirical support to merit acceptance.